# SoundNeRirF: Receiver-to-Receiver Sound Neural Room Impulse Response Field

## Abstract

We present *SoundNeRirF*, a framework that learns a continuous receiver-to-receiver neural room impulse response field (r2r-RIR) to help robot efficiently predict the sound to be heard at novel locations. It represents a room acoustic scene as a continuous 6D function, whose input is a reference receiver's 3D position and a target receiver's 3D position, and whose outputs are an inverse room impulse response (inverse-RIR) and a forward room impulse response (forward-RIR) that jointly project the sound from the reference position to the target position. Sound-NeRirF requires knowledge of neither sound source (e.g. location and number of sound sources) nor room acoustic properties (e.g. room size, geometry, materials). Instead, it merely depends on a sparse set of sound receivers' positions, as well as the recorded sound at each position. We instantiate the continuous 6D function as multi-layer perceptrons (MLP), so it is fully differentiable and continuous at any spatial position. SoundNeRirF is encouraged, during the training stage, to implicitly encode the interaction between sound sources, receivers and room acoustic properties by minimizing the discrepancy between the predicted sound and the truly heard sound at the target position. During inference, the sound at a novel position is predicted by giving a reference position and the corresponding reference sound. Extensive experiments on both synthetic and real-world datasets show SoundNeRirF is capable of predicting high-fidelity and audio-realistic sound that fully captures room reverberation characteristics, significantly outperforming existing methods in terms of accuracy and efficiency.

## 1 Introduction

Room acoustic [3, 50, 1, 27] aims at accurately capturing the sound propagation process in a room environment, so as to determine the received sound characterizing reverberation at any spatial position. Robots and other intelligent agents (e.g. voice assistants) can exploit such sound for a variety of tasks, including navigation [9, 44, 29], reconstruction [62, 8] and audio-dependent simultaneous localization and mapping (SLAM) [16]. As a complementary sensor to vision, sound perception exhibits strengths in scenarios involving dim or no lighting and occlusion. In most cases, room acoustic can be treated as linear time-invariant system (LTI). Thus, the underlying sound propagation problem is to precisely derive a room impulse response (RIR) - a transfer function of time and other acoustic variables that measures the behaviour a sound takes from a sound source to a receiver. The sound recorded at a particular position can thus be obtained by convolving the sound at the source position with the RIR. We call such RIR source-to-receiver RIR (s2r-RIR).

Accurately deriving s2r-RIR, however, is a difficult task. The challenge is three-fold: 1) high computational cost: traditional methods [3, 50, 1, 27, 41, 26, 4, 22] undergo rigorous mathematical derivation and extensive measurements to compute s2r-RIR, by treating sound as either rays [50, 1, 27] or waveforms [3, 41]. The corresponding computation complexity is proportional to the room geometric layout complexity and the number of sources. 2) non-scalable: the whole computation process needs to be re-executed once the source/receiver location changes slightly, or the room layout has altered (e.g. furniture movement). 3) too strong assumptions: the assumption that the source/receiver location and room acoustic properties are well-defined and known in advance is too strong to be held in real scenarios. The sound source location is mostly unknown in real-scenarios, and localizing sound sources is an extremely difficult task [20, 23, 6, 19].

In this work, we propose *SoundNeRirF*, a receiver-to-receiver **S**ound **N**eural **R**oom **i**mpulse **r**esponse **F**ield for efficiently predicting what sounds will be heard at arbitrary positions. SoundNeRirF requires knowledge of neither sound sources (e.g. source number and position) nor room acoustic properties[1], but instead represents a room acoustic scene through a sparse set of 3D spatial positions that robots have explored, as well as the sound recorded by the robot at each position. A robot equipped with a receiver can easily collect massive such datasets by simply moving around the room and recording its position and received sound at each step. SoundNeRirF represents a room acoustic scene by a continuous 6D function, which takes two receivers' 3D positions (one reference and one target position) as input, and outputs two room impulse responses that jointly project the sound from the reference position to the target position. SoundNeRirF thus learns receiver-to-receiver RIR (r2r-RIR). SoundNeRirF is reinforced to implicitly encode the interaction between sound sources and receivers, in addition to room acoustic properties by minimizing the error between predicted and the truly recorded sounds, because sound recorded at any position is naturally a result of such an interaction. By instantiating the 6D function as multi-layer perceptrons (MLPs), SoundNeRirF is differentiable and continuous for any spatial position, and can be optimized with gradient descent. Figure 1 visualizes SoundNeRirF's motivation.

Specifically, SoundNeRirF splits r2r-RIR into two parts: an inverse-RIR for the reference position and a forward-RIR for the target position. We further introduce receiver virtual position settings and a physical constraint strategy that guides SoundNeRirF to explicitly learn direct-path, specular-reflection and late-reverberation that commonly exist in room acoustics. SoundNeRirF has numerous applications in robotics, including immersive audio/video game experience in augmented reality (AR) [24, 57], sound pre-hearing without actually reaching to the location [5], and improving robot localization (by utilizing the predicted sound and truly heard sound).

In summary, we make four main contributions: **First**, we propose SoundNeRirF that learns r2r-RIR neural field from a sparse set of receiver recordings. **Second**, SoundNeRirF requires knowledge of neither sound source nor room acoustic properties, but instead depends on more accessible data that can be collected by robot walking around in the room environment. **Third**, SoundNeRirF directly predicts sound raw waveform. It disentangles sound prediction and r2r-RIR learning, exhibiting strong generalization capability in predicting previously unheard sound (see Experiment Sec.). **Lastly**, we release all datasets collected in either synthetic and real-world scenario to the public to foster more research.

## 2 RELATED WORK

**Room Acoustics Modelling** There are two main ways to model room acoustics: wave-based modelling [3, 41, 26, 4] and geometry-based modelling (aka geometrical acoustics) [50]. The wave-based modelling utilizes sound wave nature to model sound wave propagation, whereas geometry-based modelling treats sound propagation as optic rays. Typical geometry-based modelling methods include ray tracing [27], image source method (ISM [1]), beam tracing [17] and acoustic radiosity [22, 36]. The main goal of room acoustics is to characterize the room reverberation effect, which consists of three main components: direct-path, specular-reflection and late-reverberation. SoundNeRirF explicitly models the three components to learn r2r-RIR.

**Neural Room Impulse Response (RIR)** Some recent work [46, 55, 47, 45, 12, 30, 49] proposed to learn room impulse response (RIR) with deep neural networks. However, they all assume massive source-to-receiver RIRs (s2r-RIR) are available to train the model. Unlike these methods, Sound-NeRirF learns implicit r2r-RIR from a set of robot recorded sounds at different positions, which are easier to collect with much less constraints.

**Neural Radiance Field (NeRF)** has received lots of attention in recent years, especially in computer vision community [33, 21, 59, 52, 61]. They model static or dynamic visual scenes by optimizing an implicit neural radiance field in order to render photo-realistic novel views. Some recent work [60, 13, 58] extends to neural radiance field to 3D point cloud [60], image-text domain [58] and robotics [13, 53, 35, 28] for robot localization, position and scene representation. A. Luo *et. al.* [30] propose to learn implicit neural acoustic fields from source-receiver pairs model how nearby physical environment affect reverberation. However, their method requires the presence of massive

---

[1]room acoustic properties relate to any factor that may affect sound propagation, including room size, geometric structure, surface absorption/scattering coefficient, etc.

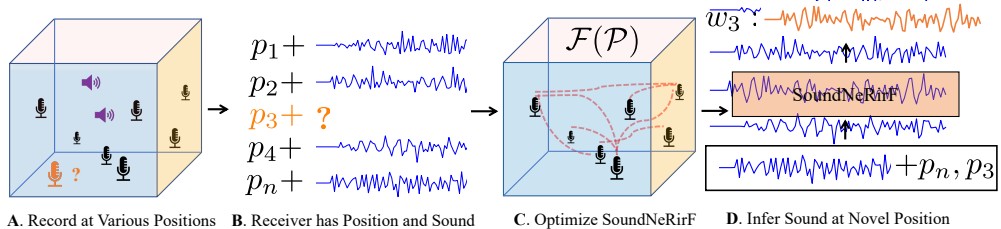

A. Record at Various Positions   **B.** Receiver has Position and Sound   **C.** Optimize SoundNeRirF   **D.** Infer Sound at Novel Position

Figure 1: SoundNeRirF motivation: in a geometric acoustic room, multiple receivers (or robot walks to different positions) at various positions are recording the sound emitted from multiple sound sources (A). Each receiver records the sound and its spatial position (B). SoundNeRirF learns receiver-to-receiver neural room impulse response field (C), so as to predict the sound at novel position, given a reference sound and a reference position (D).

s2r-RIR as well. Our proposed SoundNeRirF shares the same idea with NeRF that learns an implicit receiver-to-receiver acoustic room impulse response field.

**Sound Synthesis** Inferring sound waveform for a novel location partially relates to sound synthesis [38, 14, 64, 48, 15, 54, 11, 42]. WaveNet [38] learns to predict future sound waveform based on previously heard sound waveform. WaveGAN [14] and GANSynth [15] adopt generative adversarial network (GAN [18]) to learn to generate sound. Specifically, A. Richard *et. al.* [48] propose to infer binaural sound waveforms from mono-channel input waveform.

## 3 SOUNDNERIRF: RECEIVER-TO-RECEIVER NEURAL ROOM IMPULSE RESPONSE FIELD

### 3.1 PROBLEM DEFINITION

In a geometric acoustic room, at least one static sound source is constantly emitting sounds. $N$ sound receivers at various positions (or as a robot moves to different positions) are recording the acoustic environment, $\{(\mathcal{W}, \mathcal{P}) = \{(w_i, p_i)\}_{i=1}^N\}$, $\mathcal{W} \in \mathbb{R}^{T \times N}$, $\mathcal{P} \in \mathbb{R}^{3 \times N}$. Each sound $w_i$ is a time series of the sound raw waveform (of length $T$) and position $p_i$ is the receiver's coordinate $[x_i, y_i, z_i]$. Our target is to learn an implicit receiver-to-receiver neural room impulse response field (SoundNeRirF) $\mathcal{F}$ from $(\mathcal{W}, \mathcal{P})$, that is capable of efficiently inferring the sound $\hat{w}_t$ that would be heard at a novel position (target position) $p_t$. Note that $p_t$ is previously unexplored, and we are given a reference sound $w_r$ from explored location $p_r$,

$$\hat{w}_t = \mathcal{F}_{(\mathcal{W}, \mathcal{P})}(p_t, p_r, w_r), \quad (\hat{w}_t, p_t) \notin (\mathcal{W}, \mathcal{P}), \ (w_r, p_r) \in (\mathcal{W}, \mathcal{P}) \tag{1}$$

In this work, both sound source and receiver are omni-directional and monoaural, so we do not have to consider their directivity and inter-channel difference (as discussed in [20, 48]). We also assume all receivers record the same sound content (e.g. telephone ring) but each has encoded unique room reverberation effect based on its position, so the sound at arbitrary position can be predicted by the sound from any reference position, if the room reverberation change between them can be modelled.

### 3.2 SOUNDNERIRF FRAMEWORK INTRODUCTION

SoundNeRirF represents the receiver-to-receiver neural room impulse response field as a 6D function $h$, whose input is the combination of one reference receiver position $p_r = [x_r, y_r, z_r]$ and one target receiver position $p_t = [x_t, y_t, z_t]$, and whose output is an inverse-RIR $h^{\text{inv}}$ that mainly depends on reference position $p_r$ and a forward-RIR $h^{\text{for}}$ that mainly depends on target position $p_t$: $h^{\text{inv}}, h^{\text{for}} = h(p_r, p_t)$. $h^{\text{inv}}$ and $h^{\text{for}}$ jointly project the sound at the reference position $w_r$ to the target position $\hat{w}_t$ via 1D convolution operation,

$$\hat{w}_t = w_r \circledast h^{\text{inv}} \circledast h^{\text{for}} \tag{2}$$

where $\circledast$ indicates 1D convolution operation in time domain. Both $h^{\text{inv}}$ and $h^{\text{for}}$ are one-dimensional impulse response in the time domain, and we omit their time indexes for succinct representation. The motivation behind learning two separate RIRs is to reduce the reference-target position arbitrarity

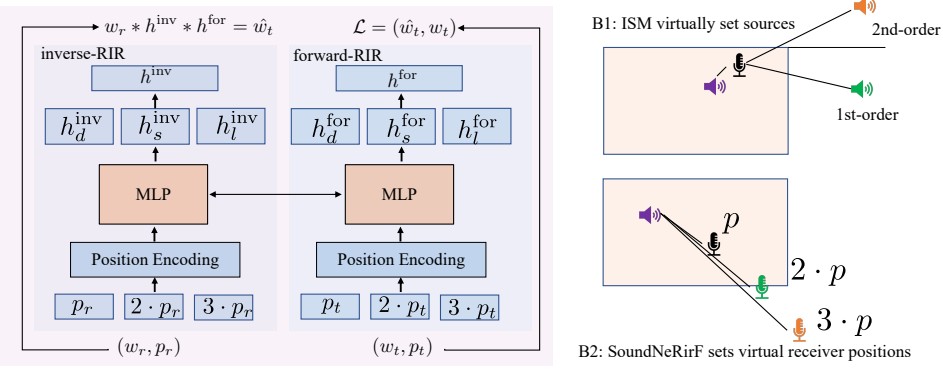

A. SoundNeRirF Pipeline Illustration        B. Virtual Receiver Setting

Figure 2: SoundNeRirF pipeline (left) and virtual receiver position setting motivation illustration.

impact and explicitly exploit room acoustic RIR property: we first project the reference sound to a virtual implicit sound source field with inverse-RIR ($h^{\mathrm{inv}}$, mainly depends on reference position) and then reproject it from the virtual implicit sound source field to the target sound with forward-RIR ($h^{\mathrm{for}}$, mainly depends on target position). By explicitly setting physical constraints to $h^{\mathrm{inv}}$ and $h^{\mathrm{for}}$ (Sec. 3.4), SoundNeRirF learns more robust r2r-RIR representation. In this work, we instantiate both $h^{\mathrm{inv}}$ and $h^{\mathrm{for}}$ as two identical multi-layer perceptrons (MLP, specifically 6 layers with 512 neurons in each layer), so they are fully differentiable and can be optimized by gradient descent algorithm. They are also continuous at any spatial position so we can infer the sound for any position,

$$h^{\mathrm{inv}} = \mathrm{MLP}^{\mathrm{inv}}(p_r), \quad h^{\mathrm{for}} = \mathrm{MLP}^{\mathrm{for}}(p_t) \tag{3}$$

MLP representation has been successfully used for acoustic tasks such as speech enhancement [37] and audio equalization [40]. We extract the last MLP layer representation as the r2r-RIR (see Fig. 2). Please note that $h^{\mathrm{inv}}$ and $h^{\mathrm{for}}$ learning are independent of the sound content, we can use them for predicting previously unheard sound (sound emitted from the same source but not trained with, see Experiment). Hitherto we have vanilla SoundNeRirF with raw receiver position as input to learn neural r2r-RIR field using an MLP. We further modify it by adding extra virtual receivers and setting physical constraints to reinforce SoundNeRirF to learn a more representative r2r-RIR neural field.

## 3.3 SETTING VIRTUAL RECEIVER POSITIONS

To explicitly model the direct-path, specular-reflection and late-reverberation effect that commonly exist in room acoustic RIR, we manipulate each original input 3D position $p$ (which holds both the reference and target receiver position) and consecutively place them at two virtual further locations (in our case, we multiply the original position by different coefficients to ensure the receiver is virtually set at further positions). The resulting three positions are leveraged to explicitly model the direct-path RIR $h_d$ (original position), specular-reflection RIR $h_s$ and the late-reverberation RIR $h_l$, respectively. The final RIR is obtained by adding three separate RIRs together: $h_d(\alpha_1 \cdot p) + h_s(\alpha_2 \cdot p) + h_l(\alpha_3 \cdot p)$, and three separate RIRs go through the same MLP. Please note that the virtual receiver position setting is adopted for both the reference position and the target position. As a result, $h^{\mathrm{inv}}$ in Eqn. 3 can be rewritten as,

$$h^{\mathrm{inv}} = h_d^{\mathrm{inv}} + h_s^{\mathrm{inv}} + h_l^{\mathrm{inv}}; \quad h_d^{\mathrm{inv}} = \mathrm{MLP}^{\mathrm{inv}}(\alpha_1 \cdot p_r), h_s^{\mathrm{inv}} = \mathrm{MLP}^{\mathrm{inv}}(\alpha_2 \cdot p_r), h_l^{\mathrm{inv}} = \mathrm{MLP}^{\mathrm{inv}}(\alpha_3 \cdot p_r) \tag{4}$$

$h^{\mathrm{for}}$ can be obtained in similar formulation in Eqn. 4. $\alpha_1, \alpha_2, \alpha_3$ are hyperparameters setting the receiver to a further virtual position (In our case, they are empirically chosen, $\alpha_1 = 1$, $\alpha_1 = 2$, $\alpha_3 = 3$). The motivation of such design is based on sound propagating as optic rays [1, 27]: the longer the path a sound signal travels, the greater the time delay and energy decay of the RIR there are in general. We implicitly approximate such room acoustic RIR nature caused by sound travelling path length difference by virtually setting the receiver at the further location. Similar approximation has been used by a traditional Imaged Sourced Method (ISM [1]), in which it mirrors the sound source against a wall to get a virtual sound source that has a different distance to the receiver. The mirroring

process can be executed one or multiple times to obtain different order virtual sources. Since we have no knowledge of the sound sources, we instead manipulate the receiver location to model the same RIR nature (See Fig. 2 right). Following the practice in [43, 63, 56, 33], we add position encoding to encode the 3D position into high-dimensional representation before feeding it to neural network.

### 3.4 Physical Constraint on r2r-RIR

To encourage SoundNeRirF to learn representative r2r-RIR, we set two constraints on MLP-learned RIR: First, time-delay order constraint for direct path $h_d$, specular-reflection $h_s$ and late-reverberation $h_l$. Second, to reflect the increased amplitude attenuation (energy decay) caused by longer travelling distance, we add monotonicity constraints to both inverse-RIR and forward-RIR.

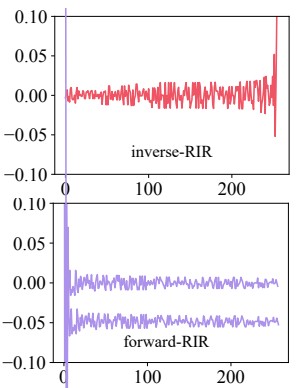

Figure 3: Learned $h^{\text{inv}}$ and $h^{\text{for}}$.

For the MLP-learned inverse-RIR $h^{\text{inv}}$ (specifically $h_d^{\text{inv}}$, $h_s^{\text{inv}}$, $h_l^{\text{inv}}$) and forward-RIR $h^{\text{for}}$ (specifically $h_d^{\text{for}}$, $h_s^{\text{for}}$, $h_l^{\text{for}}$) in Eqn. 4, we associate each individual RIR with a learnable time shift parameter ($t_d^{\text{inv}}$, $t_s^{\text{inv}}$, $t_l^{\text{inv}}$ for inverse-RIR, $t_d^{\text{for}}$, $t_s^{\text{for}}$, $t_l^{\text{for}}$ for forward-RIR). By restricting the time shift parameters in ascending order (that is, $t_d^{\text{inv}} \le t_s^{\text{inv}} \le t_l^{\text{inv}}$, $t_d^{\text{for}} \le t_s^{\text{for}} \le t_l^{\text{for}}$), we set constraints to the RIRs in Eqn. 4 to reflect different travelled distances of sound. Moreover, to reflect the greater energy decay (attenuated amplitude) caused by longer travelling distance, we add a monotonicity constraint to each learned RIR separately. For example, for the $h_d^{\text{inv}}$, the two physical constraints can be expressed as,

$$h_d^{\text{inv}}(t) = h_d^{\text{inv}}(t - t_d^{\text{inv}}); \quad h_d^{\text{inv}}(t) = \sigma\left(h_d^{\text{inv}}(t), h_d^{\text{inv}}(t-1)\right) \tag{5}$$

where $t$ is the RIR index in the time domain. The other five RIRs can be obtained in a similar way. $\sigma(\cdot)$ operation ensures monotonicity. In inverse-RIR $h^{\text{inv}}$, we use increasing monotonicity which means $\sigma = \max(\cdot)$, while in forward-RIR $h^{\text{for}}$ we use decreasing monotonicity $\sigma(\cdot) = \min(\cdot)$. Examples of typical learned inverse-RIR and forward-RIR on our synthetic dataset are shown in Fig. 3.

### 3.5 Training and Inference

The SoundNeRirF pipeline is shown in Fig. 2. It takes as input a pair of reference and target receiver positions. Two identical MLPs with layer-wise communication [34, 6] are used to learn $h^{\text{inv}}$ and $h^{\text{for}}$ in parallel. Position encoding is exploited to embed both the real and virtual positions to higher dimensions before feeding to MLP. The $h^{\text{inv}}$ and $h^{\text{for}}$ are jointly optimized by minimizing the discrepancy between the predicted sound $\hat{w}_t$ and the ground truth sound $w_t$. For the loss function, we combine mean squared error in both time and time-frequency domains,

$$\mathcal{L} = ||w_t - \hat{w}_t||^2 + ||\text{STFT}(w_t) - \text{STFT}(\hat{w}_t)||^2 \tag{6}$$

where $\text{STFT}(\cdot)$ indicates the short-time Fourier transform. We explicitly use the mean squared loss in the frequency domain because merely resorting to a mean squared loss in the time domain easily leads to audio quality issues and distorted signals [48]. During inference, theoretically we can leverage any robot traversed position and recorded sound (reference sound) to infer the sound (target sound) at an arbitrary novel location.

## 4 Experiment

**Dataset** We run experiments on three datasets:

1. **Synthetic Dataset** Single large room with 3D source/receiver positions. With the flexibility of arbitrarily setting source/receiver number or positions, we can test the necessity of each components of SoundNeRirF framework (ablation study). To this end, we adopt Pyroomacoustics [51] to simulate a large shoebox-like room of size $[50m \times 50m \times 30m]$. Seven seed sounds are collected from copy-right free website `freesound.org`: `piano`, `cat-meowing`, `footstep`, `phone-ringing`, `baby-crying`, `dog-barking` and `people-talking`. Each seed sound is 4 seconds long and with the sampling rate 16k Hz. To reflect the impact of

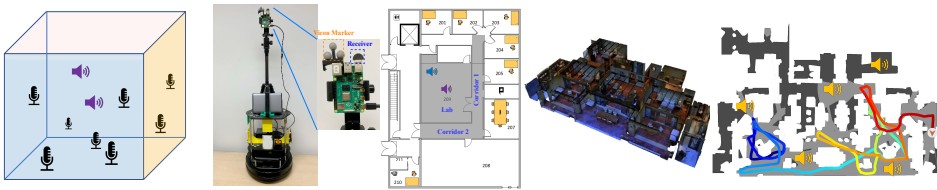

A. Synthetic Data     B. Real-World Data (left: Robot, right: Floorplan)     C. Cross-Room Data (left: 3D scene, right: robot trajectory)

Figure 4: Three datasets comparison. A. We can freely put arbitrary sources/receivers in the synthesized shoebox-like environment. B. We use Turtlebot to collect sound in lab and corridors. C. Robot explore a multiple-room scene to collect sound at various positions. The trajectory path on the topdown map evolving from blue to red color indicates the exploration chronological order.

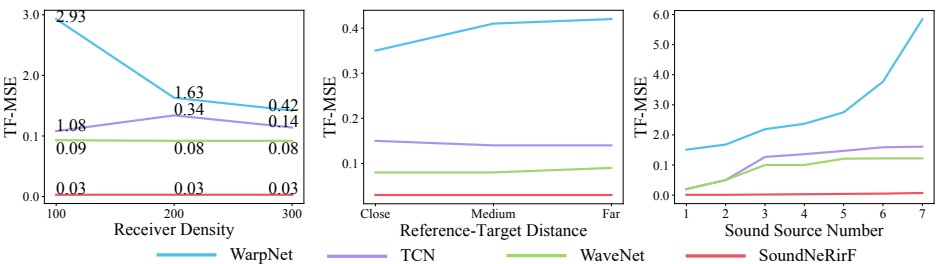

Figure 5: TF-MSE variation change w.r.t receiver density (left), reference-target distance (middle) and sound source number (right). Please note we re-scaled the vertical axis for the left/right figure for better visualization and comparison.

sound source numbers, we simulate seven identical rooms to cover sound source numbers from one to seven. In each room, we first put the corresponding sound sources at random positions, then randomly set 500 receivers at various locations (by guaranteeing at least 0.5 m distance to any sound source) to record the sound simultaneously. In each room, we keep all the sound sources and receivers' positions unchanged but replace the original seed sounds with new ones. It helps test all methods' generalization capability in predicting unheard sound. The resulting 500 recorded sounds in each room are split into 300/100/100 for train/test/eval, respectively.

2. **Real-World Dataset** Lab/Corridor environment. We use a Turtlebot robot equipped with a sound receiver (hosted by a Raspberry Pi). Vicon system [32] is utilized for localizing both speaker and receiver with a high precision (millimetre-level precision). The data-collection area consists of a cluttered lab room and two adjacent corridors. Up to 2 omni-directional speakers are put in the lab area with fixed positions. The Turtlebot moves to different positions to record sound. We collected two sub-datasets: the first one involves one sound source (emitting `baby-crying` first and then `piano`. `baby-crying` is for training model, `piano` is for unheard sound test), while the second one has two sound sources (one emitting `cat-meowing` and the other emitting `dog-barking` at the same time). In total, we collected 120 receiver positions for one and two sound sources separately, with 90/30 for train/test split.

3. **Cross-Room Dataset** large area with multiple rooms. To test the performance in room scenes that contain multiple rooms, and complex room layouts (e.g. wall, sofa and other furniture), we further run experiments on large-scale acoustic-realistic room environment platform SoundSpaces-2.0 [10]. SoundSpaces-2.0 supports Matterport3D indoor environment [7], it synthesizes high-fidelity sound (both binaural and monoaural) for any given source and receiver position. The average scene size is larger than $100m^2$. In total, we have collected 5000 sound recordings at different positions throughout the 5 room scenes (each room with 1000 recordings). We randomly split it into 700/300 for train/test, respectively. We train the individual model for each room separately. Each room has five sound sources that distribute to whole area, all sound sources emitting *telephone-ring* sound.

The comparison of the three datasets is in Table 8. More discussion about the three datasets' creation/motivation is given in Appendix A. Figure 4 shows the three datasets comparison. All datasets will be publicly released to benefit the research community.

Table 1: Overall Result on Synthetic Data

| Methods | T-MSE | TF-MSE | CDPM |
|---|---|---|---|
| LinInterp | 0.97 | 0.61 | 1.29 |
| NNeigh | 1.12 | 0.78 | 1.41 |
| WarpNet [48] | 1.17 | 0.42 | 1.28 |
| WaveNet [38] | 0.27 | 0.08 | 1.41 |
| TCN [2] | 0.43 | 0.14 | 1.45 |
| SoundNeRirF | **0.19** | **0.03** | **0.43** |

Table 2: Unheard Sound Result on Synth. Data

| Methods | T-MSE | TF-MSE | CDPM |
|---|---|---|---|
| LinInterp | 0.97 | 0.61 | 1.39 |
| NNeigh | 1.12 | 0.78 | 1.44 |
| WarpNet [48] | 1.38 | 0.66 | 1.12 |
| WaveNet [38] | 1.23 | 0.58 | 1.41 |
| TCN [2] | 1.21 | 0.76 | 1.47 |
| SoundNeRirF | **0.19** | **0.04** | **0.44** |

**Evaluation Metrics** 1) T-MSE: time domain mean squared error (T-MSE). It evaluates the difference between predicted sound and ground truth sound in the time domain; 2) TF-MSE: time-frequency mean squared error (TF-MSE). We first convert the 1D sound waveform to 2D time-frequency representation by short-time Fourier transform (STFT), and compute the mean squared error in the time-frequency domain; 3) CDPM [31, 64]: Contrastive learning-based deep perceptual audio metric. It incorporates human judgement in the loop, and is trained with a dataset carefully labelled by humans according to their perceptual similarity. We use CDPM as an approximation of human subjective evaluation of two sounds similarity. **The lower of each of the three metrics, the better of the performance.**

**Test Configuration** For the synthetic dataset, we create 50 reference sounds (selected from the training set) that cover the whole room. The sound at a test position is predicted by each of the reference sounds separately. So each test position has 50 predictions. In addition to reporting the average metrics, we rank the 50 predictions according to their distance to the test position, and divide them into three distance-aware categories: *close*, *medium* and *far*. Reporting results under the three categories helps to understand different methods' capability in predicting sound from distant, farfield positions (see Fig. 5 middle). We randomly select 30/50 reference sounds from the training set for real-world/cross-room test, respectively.

**Comparing Methods** There are no existing methods that directly model receiver-2-receiver sound prediction. We compare SoundNeRirF with five most relevant methods: two interpolation based methods and three neural network based methods (More detailed discussion is in Appendix B):

1. **LinInterp**: neighboring linear-interpolation. For each test position, we query its closest 5 positions with known sound. Then we linearly interpolate the sound to be heard at the test position by merging the 5 neighboring sound recordings with per-position weight, and each weight of the sound at a known position is inversely proportional to its distance to the test position.

2. **NNeigh**: nearest neighbor. Without any learning/interpolation process, we simply treat the nearest neighbor sound as the sound to be heard at any test (novel) position.

3. **WaveNet** [38]. The widely used WaveNet is adapted to take reference receiver position, sound and target position as input. It exploits dilated convolution to predict sound waveform.

4. **TCN** [2]. TCN adopts convolution and recurrent neural networks to model sequence2sequence transformation. We inject both the reference and target position to TCN to reinforce TCN to learn to predict target sound.

5. **WarpNet** [48]. WarpNet learns a warping field that maps the monoaural audio to binaural audio. In our implementation, we feed receiver position $[x, y, z]$ to the warping module to learn to warp the sound from the reference position to the target position.

**Training Details** We train all models with Pytorch [39] and Adam optimizer [25], with an initial learning rate 0.0005 which decays every 20 epochs. We train 100 epochs and train each model three times and report the mean value for reducing model initialization impact (standard deviations are all within 0.01, we do not report it for brevity report). All sounds have sampling rate 16 k Hz. The sound length for both synthetic and real-world data is 4 s, while for cross-room data is 1 s. All sounds are normalized to $[-1, 1]$ before feeding to the neural network for train. SoundNeRirF network architecture is given in Appendix E.

## 4.1 EXPERIMENTAL RESULT ON SYNTHETIC DATASET

The result on synthetic dataset is shown in Table 1 and Table 2 (T-MSE: $10^{-3}$). We can see that SoundNeRirF achieves the best performance over all the five comparing methods. In unheard sound

Table 3: One Sound Source Result on Real Data

| Methods | T-MSE | TF-MSE | CDPM |
|---------|-------|--------|------|
| LinInterp | 0.37 | 0.07 | 0.91 |
| NNeigh | 0.62 | 0.10 | 0.98 |
| WarpNet [48] | 0.27 | 0.06 | 0.70 |
| WaveNet [38] | 0.30 | **0.01** | 0.89 |
| TCN [2] | 0.60 | 0.02 | 0.70 |
| SoundNeRirF | **0.20** | **0.01** | **0.70** |

Table 4: Two Sound Sources Result on Real Data

| Methods | T-MSE | TF-MSE | CDPM |
|---------|-------|--------|------|
| LinInterp | 2.50 | 0.10 | 0.88 |
| NNeigh | 4.20 | 0.13 | 0.97 |
| WarpNet [48] | 3.10 | 0.10 | 0.85 |
| WaveNet [38] | 2.05 | 0.06 | 1.17 |
| TCN [2] | 2.31 | 0.07 | 0.96 |
| SoundNeRirF | **1.20** | **0.05** | **0.71** |

Table 5: Result on Cross-Room Data

| Methods | T-MSE | TF-MSE | CDPM |
|---------|-------|--------|------|
| LinInterp | 3.40 | 0.59 | 3.10 |
| NNeigh | 4.50 | 0.77 | 4.89 |
| WarpNet [48] | 3.12 | 0.30 | 2.31 |
| WaveNet [38] | 1.76 | 0.19 | 1.43 |
| TCN [2] | 2.04 | 0.26 | 1.68 |
| SoundNeRirF | **1.64** | **0.10** | **1.07** |

Table 6: Param Size/Inference Time

| Methods | Params | Inference Time | |
|---------|--------|----------------|--------|
| | | i5 CPU | Rasp. Pi 4 |
| WarpNet [48] | 8.59 M | 2.925 s | - |
| WaveNet [38] | **1.90 M** | 2.657 s | 9.377 s |
| TCN [2] | 8.57 M | 2.991 s | - |
| SoundNeRirF | 3.29 M | **0.011 s** | **0.066 s** |

test, the three learning comparing methods have observed remarkable performance drop (even worse than the two non-learning baselines), while SoundNeRirF maintains nearly the same performance and thus exhibits strong generalization capability in predicting previously unheard sound.

**Receiver Density Discussion** results reported in Table 1 and 2 are obtained by models trained on 300 receiver recordings. To figure out the impact of receiver density (number of receiver recordings), we train another two models with much smaller densities (with 100, and 200 recordings respectively). The result is shown in Fig. 5 left, from which we can see WarpNet [48] is the most sensitive to receiver density. The reduced receiver density leads to significant performance reduction. TCN [2] also suffers from low receiver density, and WaveNet [38] is less sensitive to receiver density than WarpNet and TCN. SoundNeRirF is invariant to receiver density change (TF-MSE stays as 0.03), which attests that SoundNeRirF is able to learn representative r2r-RIR with limited receiver recordings.

**Reference-Target Distance Discussion**. The TF-MSE variation w.r.t. reference-target distance is shown in Fig. 5 middle, from which we can see that the performance of WarpNet and WaveNet gradually decrease as the reference-target distance increases. SoundNeRirF and TCN are insensitive to reference-target distance, and SoundNeRirF stays as the best under all distance metrics.

**Impact of Source Number**. The performance w.r.t. sound source number is shown in Fig. 5 right, from which we can see the three comparison methods reduce in accuracy as the number of sources increases, with WarpNet [48] suffering the most. SoundNeRirF remains the best-performing method, owing to its design in learning implicit neural r2r-RIR field.

## 4.2 ABLATION STUDY

To test the necessity and effectiveness of each component of SoundNeRirF design, we test three SoundNeRirF variants: 1) SoundNeRirF without physical constraints (SoundNeRirF_noPhys), which helps test the necessity of explicitly setting physical constraints; 2) SoundNeRirF without explicitly modelling inverse-RIR and forward-RIR (SNeRirF_1MLP, instead using one MLP to

Table 7: Ablation Study Result on Synth. Data

| Methods | T-MSE | TF-MSE | CDPM |
|---------|-------|--------|------|
| SNeRirF_noPhys | 0.32 | 0.05 | 0.59 |
| SNeRirF_1MLP | 0.35 | 0.05 | 0.57 |
| SNeRirF_noVit | 0.34 | 0.05 | 0.60 |
| SoundNeRirF | **0.19** | **0.03** | **0.43** |

predict r2r-RIR); 3) SoundNeRirF without receiver virtual position setting (SNeRirF_noVit). The result is given in Table 7 (T-MSE: $10^{-3}$), from which we can see the necessity of each SoundNeRirF component. Removing each component inevitably leads to inferior performance.

| Category | Synthetic Data | Real-World Data | Cross-Room Dataset |
|---|---|---|---|
| Source Num. | 1-7 | 1-2 | 5 |
| Sound Length | 4 s | 4 s | 1 s |
| Data Size | $500 \times 7$ | 120 | $1000 \times 5$ |
| Area Size | $[50, 50, 30]$ | $[10, 6, 3]$ | $\geq 100$ |
| Train/Test | 300/100 | 90/30 | 700/300 |
| Data Distribution | 3D | 2D Plane | 2D Plane |
| Data Collector | Simulator | TurtleBot | Robot |
| Data Source | Synthetic | Real | Synthetic |

Table 8: Three Dataset Comparison

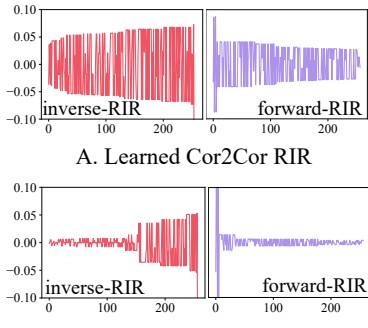

A. Learned Cor2Cor RIR

B. Learned Lab2Cor RIR

Figure 6: SoundNeRirF RIR Vis.

### 4.3 EXPERIMENTAL RESULT ON REAL-WORLD DATASET

The result on real-world dataset is given in Table 3 (one source) and Table 4 (two sources) (T-MSE: $10^{-4}$). We can learn that 1) while all methods have sharp performance drop when transiting from one sound source to two sound sources, SoundNeRirF largely mitigates the dilemma and always performs the best. Typical SoundNeRirF learned RIRs are shown in Fig. 6 and Fig. VI,V,VII in Appendix, with reference and target receiver in different positions (lab or corridor). We see that SoundNeRirF successfully characterizes the room acoustic and learns different r2r-RIRs for different positions.

**Cross-Wall Prediction Discussion** We further divide the reference and target receivers in the test set into two main categories: those in the lab room and those in the corridor, then report the performance by differentiating the reference and target receiver positions. We thus get four detailed results: Lab2Cor, Lab2Lab, Cor2Cor, and Cor2Lab (in reference-to-target format,

Table 9: TF-MSE ($\times 10^{-2}$) Result on Real Data (one source, `baby-crying`) w.r.t Lab/Cor Categories.

| Methods | Lab2Lab | Lab2Cor | Cor2Cor | Cor2Lab |
|---|---|---|---|---|
| WarpNet [48] | 3.20 | 5.18 | 10.2 | 5.09 |
| WaveNet [38] | 1.72 | 0.13 | 0.16 | 1.77 |
| TCN [2] | 1.97 | 1.08 | 2.56 | 2.14 |
| SoundNeRirF | **1.57** | **0.08** | **0.07** | **1.62** |

Lab2Cor indicates the reference receiver is in the lab room, and the target receiver is in a corridor). The result is shown in Table 9. We can see 1) SoundNeRirF outperforms all comparing methods in all Lab/Cor categories. 2) WaveNet shows advantage in Lab2Cor and Cor2Cor sound prediction, but performs poorly in the other two categories. More discussion is in Appendix Sec. C.1 and Table II.

### 4.4 EXPERIMENTAL RESULT ON CROSS-ROOM DATASET

The result on cross-room data is given in Table 5 (T-MSE: $10^{-4}$), from which we can see that SoundNeRirF stays as the best-performing method, it outperforms existing methods by a large-margin (especially the two non-learning baselines). It thus shows **SoundNeRirF generalizes well to large and complex room acoustic environment where various physical impediments, and room layout exist**. Non-learning based methods lead to the worst performance (huge performance drop), which shows received sounds largely vary w.r.t. position displacement in complex room environment.

**Inference Time/Parameter Size** result is in in Table 6. The average inference time is computed by averaging 100 independent tests (in each test, we use one reference sound to predict one target sound) on two hardwares: 1.6 GHz Intel Core i5 CPU (single core) and Raspberry Pi 4. For Raspberry Pi 4 device, we covert models to ONNX-compatible models (WarpNet [48] model and TCN [2] model cannot be converted to ONNX-compatible models). We can see **SoundNeRirF has small parameter size (just larger than WaveNet), and is extremely efficient**. We refer readers to Appendix C, D for more discussion, including learned r2r-RIR visualization, extra results and noise discussion. Appendix H to potential application discussion. In sum, SoundNeRirF exhibits superiority in predicting sound to be heard at novel positions. It generalizes well for unheard sound prediction and requires much less training data than existing methods. It can also handle physical impediment.

## 5 CONCLUSION AND LIMITATION DISCUSSION

In this work, we have proposed a receiver-to-receiver acoustic neural field for predicting the sound to be heard at novel positions, given the sound heard at known positions. We assume the acoustic scene is static and sound sources position is fixed. Moreover, we just focus on mononaural sound. One future work is to extend to binaural sound prediction.

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

APPENDIX

# A  APPENDIX 1: MORE DISCUSSION ON DATASET

## A.1  DISCUSSION ON SYNTHETIC DATASET

In the synthetic dataset, we use Pyroomacoustics [51] to simulate a shoebox-like room environment of size $[50m \times 50m \times 30m]$. The reason for creating a room of such size is two-fold:

1. A large room intrinsically involves large reverberation difference. Sound recorded by receivers at different positions thus varies dramatically, in terms of sound waveform amplitude and other room acoustic related properties. Simulating such a room better fits for testing algorithms' capability in modelling room acoustics.

2. The simulated room size is complementary to the real-world dataset environment roughly of size $[12m \times 12m \times 3m]$, which is much smaller than simulated room and robot was collecting sound data within nearly a plane (the axis pointing to sky, or $z$- axis are nearly the same). This is in contrast with the synthetic dataset in which the receivers used to collect the sound are randomly set along the $x$-, $y$- and $z$- axes, filling in the whole room.

To accurately model sound propagation process, we combine ray tracing [27] and image source method [1] to derive the sound recorded by any receiver. Combining ray tracing and image source method strikes a good balance between recorded sound accuracy and computational cost. For the room simulation, we set the origin point $[0, 0, 0]$ to be located at one corner, and all receivers' position coordinates are larger than 0.

## A.2  DISCUSSION ON REAL-WORLD DATA

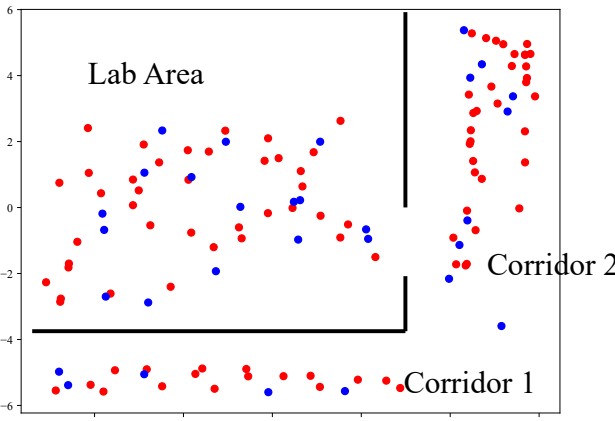

Figure I: Real-world data receiver positions visualization. Color red: positions for train; color blue: positions for test.

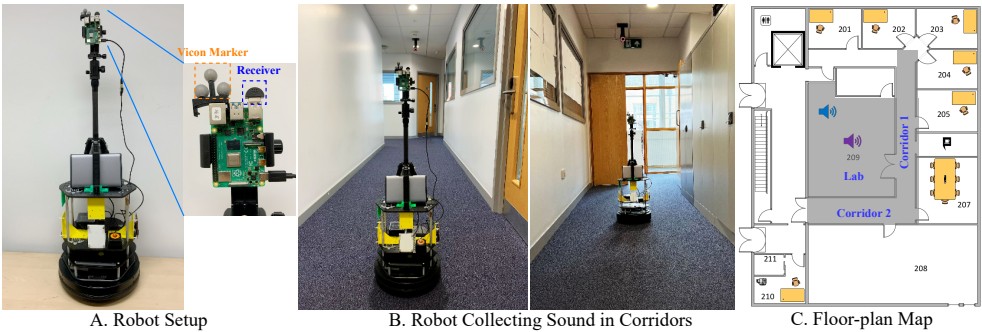

| A. Robot Setup | B. Robot Collecting Sound in Corridors | C. Floor-plan Map |

Figure II: Real-world data collection setup.

In the real-world environment, we have collected 65 receiver recordings in the two adjacent corridors, and 55 receiver recordings in the lab environment. These receiver locations cover all navigable area in the lab and two corridors. The train/test split is randomly chosen and uniformly covers both the lab and two corridors. In the real-world dataset, the $x$- axis range is $[-2.0\,m, 4.0\,m]$, $y$- axis range is $[-6.0\,m, 6.0\,m]$, $z$- range is $[0.0\,m, 0.3\,m]$. The plane formed by $x$- and $y$- axes is the 2D floor-plan map as shown in the main paper.

The TurtleBot sound receiver positions are shown in Fig. I and detailed setup is in Fig. II, in which we can observe that the positions TurtleBot has traversed uniformly cover both the lab area and two adjacent corridors. The receivers used to train (color red) and test (color test) also cover the lab area and two corridors.

## A.3 Discussion on Cross-Room Data

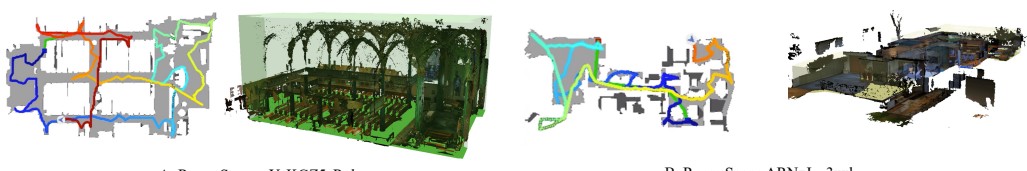

A: Room Scene gYvKGZ5eRqb      B: Room Scene ARNzJeq3xxb

Figure III: Embodied robot collects sound in complex room environment (figure A) and multiple rooms (figure B). The left sub-figure in each figure shows the robot exploration trajectory in the top-down view, the trajectory evolving from blue to red color indicates the exploration chronological order. The right sub-figure visualizes the 3D scene environment.

To test the performance in room scenes that contain multiple rooms, complex room layout (e.g. room geometric features, sofa, bed and other furniture), we further run experiments on large-scale acoustic-realistic room environment platform SoundSpaces-2.0 [10]. SoundSpaces-2.0 supports Matterport3D photo-realistic indoor environment [7], it synthesizes high-fidelity sound (both binaural and monoaural) for any given source position and receiver position. In our setting, we leverage the embodied robot to efficiently explore 5 room scenes. The robot step size is $1.0\,m$ and turn-angle is $30°$, it contains three actions: `move-forward`, `turn-left` and `turn-right`. At each time step, it records a mono-channel sound that is 1 second long sound. Each room contain five sound sources that across the whole area, both emitting `telephone-ring` sound. Figure III shows two scene examples and embodied robot exploration trajectory, from which we can see Matterport 3D room scenes represent real-world large indoor room environment where cluttered physical impediments and complex room layout exist. In total, we have collected 5000 sound recordings at different positions throughout the 5 room scenes (each room with 1000 recordings). For the 1000 recordings in each room, we randomly split it into 700 for train and 300 for test, respectively. We adopt the same training strategy to the all models, as we do for the other two datasets.

## B Appendix 2: More Discussion on Comparing Methods

### B.1 Two Non-Learning Baselines

In our experimental setting, we have a set of receivers' recordings that distribute a room area. We use these known recordings to predict the sound to be heard at novel positions. Two simple baselines are just interpolating the sound at novel position from a subset of receiver recordings with known sound and positions. Here we introduce two interpolation based baselines:

- neighboring linear-interpolation (**LinInterp**): for each test position (novel position), we query its closest 5 positions with known sound. Then we linearly interpolate the sound to be heard at the test position by merging the 5 neighboring sound recordings with per-position weight, each weight of the sound at a known position is inversely proportional to its distance to the test position and all weights sums to 1 after `softmax` constraint.
- nearest neighbor (**NNeigh**): without any learning or interpolation process, we simply treat the nearest neighbor sound as the sound to be heard at any test (novel) position. This baseline

Table I: WaveNet comparison with one SoundNeRirF variant (SoundNeRirF_1MLP) on Synthetic Dataset over sound source number from 1 to 7.

| Methods | T-MSE $10^{-3}$($\downarrow$) | TF-MSE ($\downarrow$) | CDPM ($\downarrow$) | Params. |
|---|---|---|---|---|
| WaveNet [38] | **0.27**±0.00 | 0.08±0.01 | 1.41±0.01 | 1.90 M |
| SoundNeRirF_1MLP | 0.35±0.00 | **0.05**±0.01 | **0.57**±0.01 | **1.64 M** |

helps to test if deep neural network based methods truly learn meaningful representation to sound rendering.

From the results on the three datasets, we can learn that 1) The two non-learning based baselines (LinInterp and NNeigh) lead to inferior performance than the other four deep neural network based methods (three comparing methods and our SoundNeRirF); 2) LinInterp performs better than NNeigh, which shows incorporating neighboring sounds can better capture the sound characteristic at a novel position. In summary, we can conclude that: **explicit learning process is essential for accurate room sound rendering, it helps to better characterize room acoustics**.

### B.2 Three Learning-Based Methods

There are no existing methods that directly model receiver-to-receiver sound neural field. We thus compare SoundNeRirF with three relevant methods: WarpNet [48], TCN [2] and WaveNet [38]. WarpNet [48] synthesizes binaural audio from monoaural audio. It learns a warping field that maps the monoaural audio to binaural audio. In our implementation, we feed receiver position $[x, y, z]$ to warping module to learn the warp field. TCN [2] adopts convolution and recurrent neural network to model sequence2sequence transform. WaveNet [38] adopts dilated convolution to predict future sound, based on previous heard sound. In our implementation of TCN [2] and WaveNet [38], we concatenate receiver position $[x, y, z]$ to the input sound waveform and feed their concatenation to neural network. Please note that normalization of the input receiver position $[x, y, z]$ is applied as well, as we do in SoundNeRirF.

The comparison of learnable parameter number among these methods are shown in Table 6. We can see from table that SoundNeRirF parameter number is much smaller than those of TCN [2] and WarpNet [48]. SoundNeRirF parameter number is larger than the parameter number of WaveNet [38], mainly due to the two paralleling MLPs for modelling inverse-RIR and forward-RIR respectively and layerwise feature communication strategy. We find that SoundNeRirF without learning reverse-RIR and forward-RIR respectively, but instead use one MLP to learn one general RIR (which means we just use 6-layer MLP to learn r2r-RIR, we call it SoundNeRirF_1MLP) achieves comparable and even better performance than WaveNet [38] on the synthetic dataset. The result is shown in Table I, from which we can see SoundNeRirF_1MLP outperforms WaveNet [38] in terms of TF-MSE and CDPM metrics, with a much smaller parameter number. It thus shows the advantage of SoundNeRirF in modelling receiver-to-receiver sound neural field.

The average inference time of the comparing three methods and SoundNeRirF is given in Table 6. The average inference time is computed by averaging 100 independent tests (in each test, we use one reference sound to predict one target sound) on two hardwares: 1.6 GHz Intel Core i5 CPU (single core) and Raspberry Pi 4. For Raspberry Pi 4 device, we covert trained models to ONNX-compatible models and run ONNX models on the device. We find that WarpNet [48] model and TCN [2] model cannot be converted to ONNX-compatible models successfully due to various issues, such as operator incompatibility and unsupported operators. Therefore, we only report the inference time of WaveNet [38] and SoundNeRirF. From Table 6, we can clearly see that SoundNeRirF's inference time is much shorter than the three comparing methods' inference time, with just 1/200 of the three comparing methods' inference time on CPU device, 1/150 of WaveNet [38] inference time on Raspberry Pi 4 (although SoundNeRirF has more trainable parameters than WaveNet). Therefore, **it shows the high efficiency of SoundNeRirF**.

## C  APPENDIX 3: MORE EXPERIMENTAL RESULTS

### C.1  TWO NON-LEARNING BASELINES ON CROSS LAB-CORRIDOR EXPERIMENT

We note that the neighboring sound recordings collected for the two non-learning based baselines lie very close to the test position. As a result, the sound recordings used to interpolate and the sampled nearest neighbor sound spatially lie in the same local room environment with the sound to be predicted (e.g. both lie in the lab environment or corridor environment, see Fig. I). To better test the two non-learning baselines' capability in predicting sound of further distance, we specifically compartment all sound recordings with known positions into different room environments (e.g. lab and corridor), and just use sound recordings in one environment to predict the sound in another room environment. The result (under TF-MSE evaluation metrics) is shown in Table II, from which we can see that the performance of LinInterp and NNeigh deteriorates significantly when predicting sound in one room environment by sound recordings from another different room environment (the largest TF-MSE increase by Lab2Cor and Cor2Lab). It thus shows **explicitly learning process is required to implicitly encoding room acoustic properties**.

Table II: TF-MSE ($\times 10^{-2}$) Result on Real Data (one source, `baby-crying`) w.r.t Lab/Cor Categories.

| Methods | Lab2Lab | Lab2Cor | Cor2Cor | Cor2Lab |
|---|---|---|---|---|
| LinInterp | 4.10 | 8.21 | 12.1 | 9.23 |
| NNeigh | 4.81 | 9.17 | 13.0 | 10.67 |
| WarpNet [48] | 3.20 | 5.18 | 10.2 | 5.09 |
| WaveNet [38] | 1.72 | 0.13 | 0.16 | 1.77 |
| TCN [2] | 1.97 | 1.08 | 2.56 | 2.14 |
| SoundNeRirF | **1.57** | **0.08** | **0.07** | **1.62** |

### C.2  MORE QUALITATIVE RESULTS, LEARNED R2R-RIR VISUALIZATION

We first provide two learned r2r-RIR visualizations on MP3D dataset [7]. We provide one corridor-to-corridor (Cor2Cor, both reference and target position are in corridor) learned receiver-to-receiver RIR in Fig. IV. From this figure, we can clearly see that in large spatial area MP3D dataset [7] where the average area is larger than $100\ m^2$, the direct-path RIR, specular-reflection RIR and late-reverberation RIR learn different time shift parameters. Such different time-shift parameters characterize room acoustics properties in large spatial area where the direct-path sound arrives to a receiver position earlier than specular-reflection sound, specular sound arrives earilier than later-reverberation sound.

We also provide corridor-to-corridor (Cor2Cor, both reference position and target position are in corridor) learned receiver-to-receiver RIR in Fig. V, lab-to-corridor (Lab2Cor, reference position in lab, target position are in corridor) learned receiver-to-receiver RIR in Fig. VI. lab-to-lab (Lab2Lab, both reference and target position are in lab) learned receiver-to-receiver RIR in Fig. VII. From the three figures, we can see that in much smaller area in our real-world dataset, the learned time-shift difference between direct-path, specular-reflection and later-reverberation is much smaller. Such phenomena is compatible with real-scenario room acoustics property which shows that in small area the arrival time for direct-path, specular-reflection and late-reverberation is very close.

In summary, from such learned r2r-RIR visualizations, we can conclude that SoundNeRirF learned RIR is capable of characterizing 3D room scene acoustically, from the receiver-to-receiver modelling perspective.

### C.3  MORE ABLATION STUDY

In the main paper, we have tested three SoundNeRirF variants: 1) SoundNeRirF without physical constraints (SoundNeRirF_noPhys), which helps test the necessity of explicitly setting physical constraints; 2) SoundNeRirF without explicitly modelling inverse-RIR and forward-RIR (SNeR-irF_1MLP, instead using one MLP to predict r2r-RIR. It achives slightly better performance than WaveNet [38]); 3) SoundNeRirF without receiver virtual position setting (SNeRirF_noVit). The

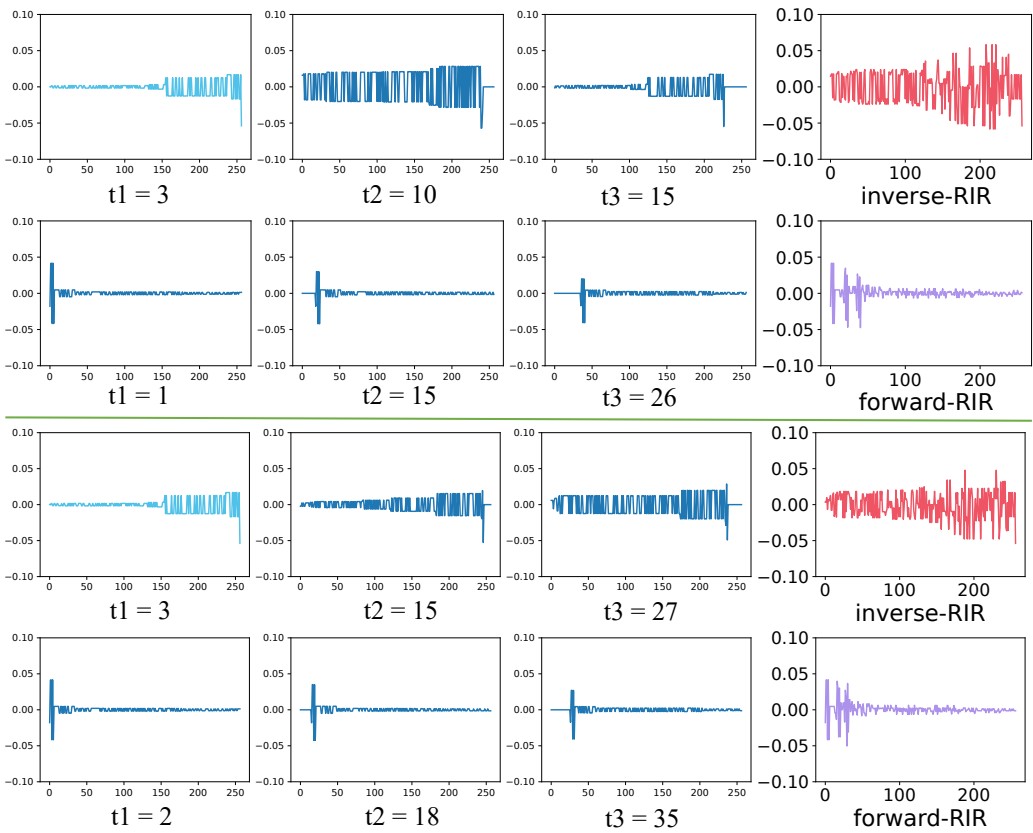

Figure IV: Two SoundNeRirF learned r2r-RIRs on MP3D dataset [7], we show two r2r-RIRs (split by the green line), each comprises of a forward-RIR and an inverse-RIR. We explicitly show the time-shift learned by the each direct-path, specular-reflection and later-reverberation.

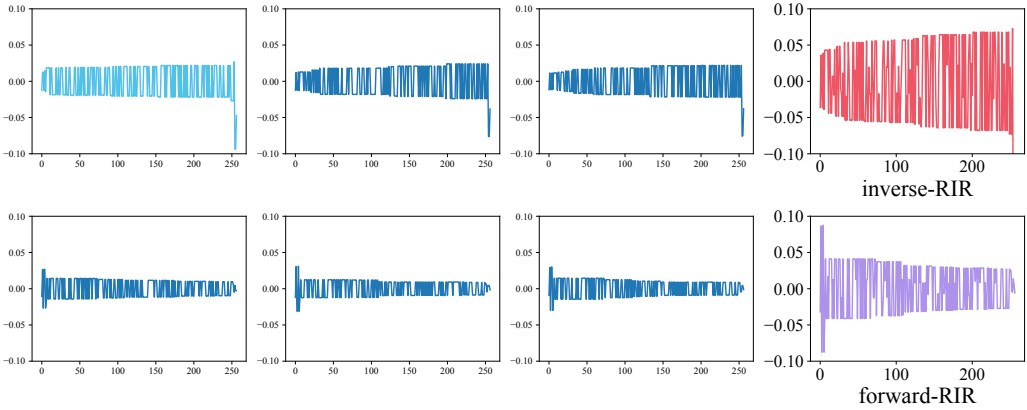

Figure V: SoundNeRirF learned corridor-to-corridor RIR. The left three sub-figures are learned specular, early and late reverberation RIR respectively.

corresponding result on synthetic dataset shows the necessity and importance of each compoment of SoundNeRirF design.

In addition to the three ablation studies, we further conduct three more studies to figure out the necessity of setting different monotonicity constraints to inverse-RIR and forward-RIR respectively. Specifically, the necessity of increasing monotonicity to inverse-RIR and decreasing monotonicity

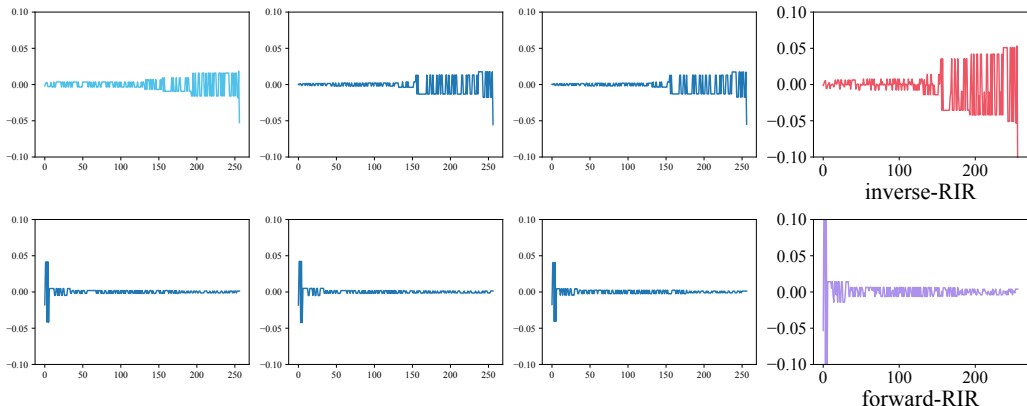

Figure VI: SoundNeRirF learned lab-to-corridor RIR. The left three sub-figures are learned specular, early and late reverberation RIR respectively.

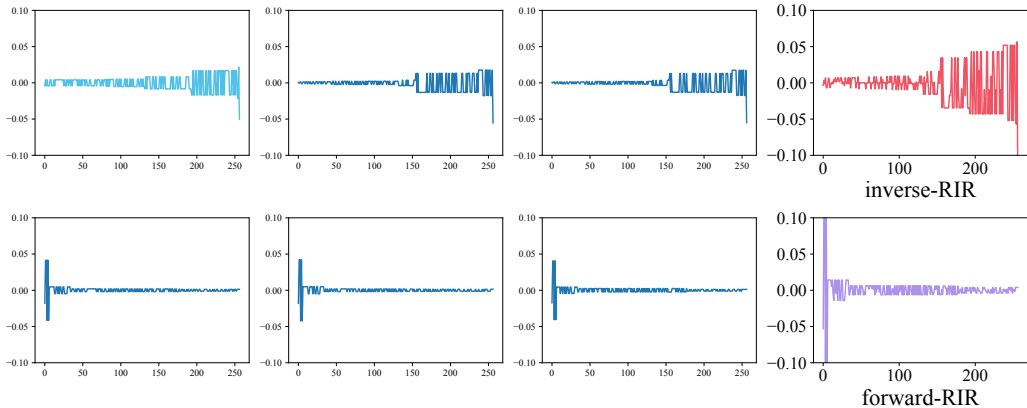

Figure VII: SoundNeRirF learned lab-to-lab RIR. The left three sub-figures are learned specular, early and late reverberation RIR respectively.

to forward-RIR. We test two SoundNeRirF variants: both inverse-RIR and forward-RIR are with increasing monotonicity constraints (SoundNeRirF_incmono), with decreasing monotonicity constraints (SoundNeRirF_decmono), and swap the original constraint (SoundNeRirF_swapmono, use decreasing monotonicity for inverse-RIR and increasing monotonicity for forward-RIR). The result is given in Table III. We can conclude by this table that applying the same monotonicity constraints (either increasing or decreasing) to both inverse-RIR and forward-RIR leads to inferior performance. Swapping the monotonicity generates the worst performance. It in turn shows the necessity of setting increasing monotonicity to inverse-RIR and decreasing monotonicity to forward-RIR.

## C.4 MORE EXPERIMENT IN REAL-WORLD DATASET

In the main paper, we have reported the quantitative result on both one sound source and two sound sources setting. The result on unheard sound test is given in Table IV. From this table, we can see that SoundNeRirF performs the best for unheard sound prediction in real-world setting.

## C.5 NOISE DISCUSSION

In real-scenario, robot position localization often suffers from noise, in which the robot cannot localize itself with high-precision. In our real-world sound data collection platform, Vicon localization system [32] gives very high-precision position coordinates of the robot at each position. To test the

Table III: Ablation Study on Monotonicity test

| Methods | T-MSE $10^{-3}$ ($\downarrow$) | TF-MSE ($\downarrow$) | CDPM ($\downarrow$) |
|---|---|---|---|
| SoundNeRirF_incmono | 0.30±0.00 | 0.05±0.01 | 0.47±0.00 |
| SoundNeRirF_decmono | 0.31±0.00 | 0.05±0.01 | 0.48±0.01 |
| SoundNeRirF_swapmono | 0.37±0.00 | 0.07±0.01 | 0.52±0.01 |
| SoundNeRirF | **0.19**±0.00 | **0.03**±0.01 | **0.43**±0.01 |

Table IV: Unheard Sound Test on Real-World Dataset

| LinInterp | 0.37 | 0.07 | 0.93 |
|---|---|---|---|
| NNeigh | 0.62 | 0.10 | 1.02 |
| Methods | T-MSE $10^{-3}$ ($\downarrow$) | TF-MSE ($\downarrow$) | CDPM ($\downarrow$) |
| WarpNet [48] | 0.17±0.00 | 0.06±0.01 | 0.93±0.01 |
| WaveNet [38] | 0.03±0.01 | **0.01**±0.01 | 0.89±0.01 |
| TCN [2] | 0.06±0.01 | 0.02±0.01 | 0.82±0.01 |
| SoundNeRirF | **0.02**±0.00 | **0.01**±0.01 | **0.79**±0.01 |

robustness of SoundNeRirF towards position noise interference, we specifically add noise pollution to Vicon provided position coordinates and test all models' performance over position noise interference.

Specifically, we model noise as Gaussian noise with mean 0 and standard deviation $\sigma$, $\mathcal{G}(\mu = 0, \sigma)$. Each coordinate $x$, $y$ or $z$ is added a noise sampled from the Gaussian noise. By varying the standard deviation $\sigma$, we test all models' tolerance w.r.t. different level of noise interference. In this experiment, we choose three noise levels, with $\sigma = 0.5, 1.0, 1.5$, respectively. The result is shown in Table V, from which we can see that SoundNeRirF exhibits stronger rubustness towards noise interference than three comparing methods.

## D  MORE DISCUSSION ON EXPERIMENTAL SETTING

For each receiver position $[x, y, z]$, we first normalize each coordinate into $[-1, 1]$ before feeding to position encoding module, as WarpNet [48] does. In position encoding, the corresponding position encoding embedding size is 384. The learned time shift parameters for both the inverse-RIR and forward-RIR are quite small (both are less than 5).

**Unheard Sound Test Explanation**. In our experimental settings, in each sound source and receiver positions, we collect two sounds. While one sound is used for training the model, the other sound is hidden for training but instead is used for testing. Since the two sounds theoretically share the same receiver-to-receiver RIR, such "unheard sound test" helps test each model's generalization capability in predicting new sound.

**Position Encoding** since deep neural networks tend to learn lower frequency functions [43], we follow the practice in 3D protein structure modelling [63], Transformer network [56] and RGB image based neural radiance field [33] to encode the the low-dimensional position into high dimension. For example, for the reference position $p_r$ (also for the two virtually set positions $2 \cdot p_r$ and $3 \cdot p_r$), the position encoding process works as,

$$\text{PosEncode}(p_r) = [\sin(2^0 \pi p_r), \cos(2^0 \pi p_r), \cdots, \sin(2^{L-1} \pi p_r), \cos(2^{L-1} \pi p_r)] \tag{7}$$

where $p_r = [x_r, y_r, z_r]$, and $L$ is the position encoding length, we use 64 in our work (so the position embedding size is 384). The target position $p_t$ goes the same process.

Table V: Noise Test on Real Data. In each evaluation metric entry, from left to right the four values correspond to metrics with noise standard deviation $\sigma = 0.0, 0.5, 1.0, 1.5$ respectively. The standard deviation for each metric is within 0.02.

| Methods | T-MSE ($10^{-4}$) ($\downarrow$) | | | | TF-MSE ($\downarrow$) | | | | CDPM ($\downarrow$) | | | |
|---------|------|------|------|------|------|------|------|------|------|------|------|------|
| WarpNet [48] | 3.10 | 3.10 | 3.17 | 3.25 | 0.10 | 0.13 | 0.15 | 0.17 | 0.85 | 0.87 | 0.88 | 0.89 |
| WaveNet [38] | 2.05 | 2.07 | 2.09 | 2.12 | 0.06 | 0.07 | 0.09 | 0.11 | 1.17 | 1.18 | 1.19 | 1.19 |
| TCN [2] | 2.31 | 2.31 | 2.32 | 2.33 | 0.07 | 0.08 | 0.08 | 0.10 | 0.96 | 0.96 | 0.98 | 0.99 |
| SoundNeRirF | **1.20** | **1.20** | **1.21** | **1.22** | **0.05** | **0.05** | **0.06** | **0.06** | **0.71** | **0.72** | **0.72** | **0.74** |

Table VI: SoundNeRirF Network Architecture. The format FC@n1,n2 indicates a full-connection layer with input neurons n1 and output neurons n2. Batch normalization (BN) is applied after each full-connection layer.

| Reference Position MLP | Target Position MLP |
|------------------------|---------------------|
| **Input:** Ref. Pos. [B, 3] | **Input:** Target Pos. [B, 3] |
| Position Encoding | |
| [B, 384] | [B, 384] |
| FC@384, 512 | FC@384, 512 |
| Layer1 Communication | |
| FC@512, 512 | FC@512, 512 |
| Layer2 Communication | |
| FC@512, 512 | FC@512, 512 |
| Layer3 Communication | |
| FC@512, 512 | FC@512, 512 |
| Layer4 Communication | |
| FC@512, 512 | FC@512, 512 |
| Layer5 Communication | |
| FC@512, 512 | FC@512, 512 |
| time-shift head: FC@512, 1 | time-shift head: FC@512, 1 |
| RIR head: FC@512, 257 | RIR head FC@512, 257 |
| **Output:** inverse-RIR [B, 257] | **Output:** forward-RIR [B, 257] |

## E  SOUNDNERIRF NETWORK ARCHITECTURE

SoundNeRirF network architecture is given in Table VI. In the layerwise communication, a trainable weight $w$ is used to merge the intermediate features arising from reference position MLP (e.g. $f_1$) and target position MLP (e.g. $f_2$) before feeding them to the next layer, respectively,

$$[f_1; f_2] = w \cdot [f_1; f_2] \tag{8}$$

## F  EXPLANATION ON FIG. 5 IN MAIN PAPER

Table VII: TF-MSE ($\downarrow$) variation change w.r.t. Reference-Target Distance (corresponds to Fig. 5 middle figure in Fig. 5 main paper)

| Methods | Close | Medium | Far |
|---------|-------|--------|-----|
| WarpNet [48] | 0.35 | 0.41 | 0.42 |
| WaveNet [38] | 0.08 | 0.08 | 0.09 |
| TCN [2] | 0.15 | 0.14 | 0.14 |
| SoundNeRirF | **0.03** | **0.03** | **0.03** |

In Fig. 5 of the main paper, the curves in the middle and right figures are not associated with the exact value, and we re-scaled the $y$- axis for better comparison. To present more quantitative comparison, we provide the exact value for each curve. The result of TF-MSE variation change w.r.t. reference-target distance is given in Table VII. The result of TF-MSE variation change w.r.t. sound source number is given in Table VIII.

## G    PREDICTED SOUND SAMPLES

We also provide predicted sound samples by each method with our real-world dataset, as well as the ground truth sound in the supplementary material. The predicted sound samples use the reference sound in the lab area and target sound in the corridor area.

Table VIII: TF-MSE ($\downarrow$) variation change w.r.t. Sound Source Number (corresponds to Fig. 5 right figure in Fig. 5 main paper)

| Method | Sound Source Number | | | | | | |
|---|---|---|---|---|---|---|---|
| | 1 | 2 | 3 | 4 | 5 | 6 | 7 |
| WarpNet [48] | 0.91 | 1.18 | 1.69 | 1.87 | 2.25 | 3.77 | 5.84 |
| WaveNet [38] | 0.02 | 0.05 | 0.09 | 0.09 | 0.11 | 0.12 | 0.12 |
| TCN [2] | 0.02 | 0.05 | 0.14 | 0.26 | 0.27 | 0.29 | 0.31 |
| SoundNeRirF | **0.01** | **0.01** | **0.02** | **0.03** | **0.04** | **0.05** | **0.07** |

## H    POTENTIAL SOUNDNERIRF APPLICATION IN ROBOTICS

The motivation of this work is to enable the robot to efficiently predict the sound to be heard at novel positions, without reply on enough prior information about the room geometric features/sound source installation and rigorous and complex mathematical derivation. The robot can efficiently and effectively learn such neural acoustic field by simply exploring a new environment to record the position and received sound at each step.

We further show that learning such sound neural field through our proposed SoundNeRirF framework is easy to implement on a robot. Our experiment shows our method just needs a relatively small number of receiver recordings to learn an efficient sound neural field (Fig. 5, left-most figure). So the robot doesn't have to walk a long time (exhaustive walking). At the same time, SoundNeRirF is light-weight, it achieves real-time inference even on resource-constrained edge device. For example, in the Table 6, SoundNeRirF takes 0.066s to predict the sound on Raspberry Pi. This advantage enables the robot to predict/render sound in real-time, so it fits for tasks that lay much emphasis on efficiency.

We argue that the learned SoundNeRirF can be used for various down-stream robot-related tasks. As far as we can see at current moment, it includes but is not limited to: 1. SoundNeRirF can be used to improve robot self-localization capability, by comparing the predicted sound and truly recorded sound at a particular position; 2. SoundNeRirF can be used for embodied robotics research in synthesized audio-vision environment. For example, the audio-vision platform for embodied robotics SoundSpaces1.0 just provides discrete and pre-computed RIR, so the whole environment is not dynamic and the robot can only hear sound at particular positions. SoundNeRirF can help to avoid such obstacles, enabling the robot to render sound at arbitrary positions. 3. VR/AR applications for immersive acoustic experience (audio auralisation).

