# OpenReview forum: "SoundNeRirF: Receiver-to-Receiver Sound Neural Room Impulse Response Field"
_ICLR.cc/2023/Conference — Submitted to ICLR 2023_

### Official Review · Reviewer_cCRH · 2022-10-22

**Confidence:** 5
**Correctness:** 4
**Technical Novelty And Significance:** 3
**Empirical Novelty And Significance:** 3
**Recommendation:** 6

**Clarity, Quality, Novelty And Reproducibility:**

The method is clear and novel. The paper seems to be easy to follow if the dataset is released.

**Strength And Weaknesses:**

Strength:
1. The paper solves the RIR prediction in a novel view: modeling the sound field with some positions with acoustic information, which is somewhat a interpolation method. Receiver-to-receiver sound field modeling requires no knowledge about the source and can support more complex scene.
2. The author decomposes the receiver-to-receiver RIR prediction into two parts: inverse-RIR and forward RIR, which is a very interesting design and has physical intuition.
3. The experiments are convincing and adequate, consisting of several datasets with different source number. The ablation verifies the effectiveness of each proposed technique.

Weakness:
1. Since the impulse response is the response of an impulse, it is a time-delay function (i.e., the impulse response assumes the impulse is emitted at the time 0, and the receiver gets the signal at positive (delay) timestamp). When we predict the waveform for a new position p_new, we convolve the waveform of a known position p_known with two IRs (inverse-RIR and forward RIR). In this case, the waveform will be further delayed, which means the method cannot handle the case that p_new is closer to the source than p_known. Maybe there are some tricks on applying the convolution?
2. In the section 3.4 (second paragraph), it seems that some subscripts are wrong (e.g., t^inv_d < t^inv_d < t^inv_l), which makes the physical constraint hard to understand. Can you state it more clearly?
3. Although the space limitation can be a reason, I think it is necessary for a paper to have a conclusion.
4. Just a question, will the dataset used in the paper be released to make the future work on this direction easier?


**Summary Of The Paper:**

The author proposes a receiver-to-receiver sound neural room acoustics rendering field, which is different with previous works focusing on source-to-receiver RIR prediction. To be specific, the author decomposes the receiver-to-receiver RIR prediction into two parts: inverse-RIR and forward RIR, which is a very interesting design.

**Summary Of The Review:**

The paper is novel and solid. However, there are still some missed details that can be further clarified.

---

> ### Author Response · Authors · 2022-11-17
> **Feedback to Reviewer #cCRH**
>
> We sincerely thank you for your constructive comments, and agreeing with the contribution made by our work. For your several raised concerns, we give one by one feedback.
>
> **Q1:** Since the impulse response is the response of an impulse, it is a time-delay function (i.e., the impulse response assumes the impulse is emitted at the time 0, and the receiver gets the signal at positive (delay) timestamp). When we predict the waveform for a new position p_new, we convolve the waveform of a known position p_known with two IRs (inverse-RIR and forward RIR). In this case, the waveform will be further delayed, which means the method cannot handle the case that p_new is closer to the source than p_known. Maybe there are some tricks on applying the convolution?
>
> **A1:** Thanks for pointing this out.
>
> 1. You are right that the received sound recordings by receivers at different positions have a time-delay difference between each other, if all receivers begin to record the sound at the same time.  However, in our case, we focus on “what does the audio sound like at a novel position, given the heard sound at another position”. So time-delay caused by the location of sound source and receiver is not the main focus of this paper. In other words, the start-time/end-time for each receiver’s recording is synchronized in the time axis.
>
> 2. If we consider the time-delay between different receivers, extra challenges must be handled. For example, the received sound length of different receivers will be different and some silent sound clip may exist, such a challenge requires special 1D convolution to process. Moreover, the time-delay directly relates to the source-receiver position, since in our work we assume we don’t have prior knowledge about source position, it requires more exploration to solve the challenge of how to implicitly involve time-delay without any sound source information. Since our work is the first to model receiver-to-receiver neural field, we hope to motivate more research in future.
>
>
> **Q2:** In the section 3.4 (second paragraph), it seems that some subscripts are wrong (e.g., $t^{inv}_d < t^{inv}_d < t^{inv}_l$), which makes the physical constraint hard to understand. Can you state it more clearly?
>
> **A2:** We are sorry for the unclear presentation. The right expression is $t^{inv}_d < t^{inv}_s < t^{inv}_l$, which means, in inverse-RIR the time-shift for direct-path is smaller than than the time-shift for specular reflection, time-shift for specular reflection is further smaller than late-reverberation time-shift . The same constraint applies to forward-RIR as well. We set such constraint to explicitly model room acoustic effects of direct-path (which comes first), specular reflection (which comes second) and late reverberation (which comes last).
>
> We have revised it in our updated version. Again we thank you for your comment on this.
>
> **Q3:** Although the space limitation can be a reason, I think it is necessary for a paper to have a conclusion.
>
> **A3:** Thanks for pointing out this. We have included the “Limitation Discussion” section at the end of the main paper. We will incorporate the conclusion into it in our updated version.
>
> **Q4:** Just a question, will the dataset used in the paper be released to make the future work on this direction easier?
>
> **A4:**  Thanks for mentioning this. As we have said in the main paper, the dataset will be publicly released to foster more research.

---

> > ### Comment · Reviewer_cCRH · 2022-11-23
> > **Thanks for your feedback**
> >
> > I would like to thank the authors for feedback to my review comments. I would like to retain my scores as before.

---

### Official Review · Reviewer_tmuv · 2022-10-24

**Confidence:** 4
**Clarity, Quality, Novelty And Reproducibility:** 1.The authors propose to release the …
**Correctness:** 3
**Technical Novelty And Significance:** 3
**Empirical Novelty And Significance:** 3
**Recommendation:** 6

**Strength And Weaknesses:**

Strengths
1. Evaluation across a variety of datasets.
2. Proposed method significantly outperforms comparative baselines.
3. Promised release of the referenced datasets.


Growth Opportunities
1. I don't understand this statement. "We also assume all receivers record the same sound content, so the sound at any target position can be inferred by the sound from any reference position." Isn't the sound received a function of the location and the propagation characteristics of the room?
2. The motivation of using specifically 2 virtual receiver positions in 3.2 is unclear.
3.  The drawbacks of classical RIR estimators is misleading as the proposed approach required significant number of labeled pairs for training, a key limitation.
4. No justification is provided for the design of the specific MLP in Section 3.2 is provided. It would be also helpful to understand how sensitive the solution is to the architecture.
5. Results of figure 5.a is confusing. If the proposed approach is truly independent of the number of receiver positions, it would be worth establishing how low the number could be.
6. Additional testing on other room sizes and configurations would be helpful, especially with respect to different absorbent/reflection surfaces.

**Summary Of The Paper:**

The authors propose a supervised learning approach to learning the room impulse response function, specifically by leveraging multi-layer perceptron (MLP). Evaluation on a range of datasets shows that the proposed approach outperforms several recent approaches.

**Summary Of The Review:**

The paper proposes a novel approach of learning room impulse response by leveraging a mobile robot to collect additional samples of the same source(s). Writing and clarity could be strengthened as well as addressing some of the aspects highlighted above to make it a strong paper

---

> ### Author Response · Authors · 2022-11-16
> **Feedback to Reviewer #tmuv**
>
> We sincerely thank you for your constructive comments and feedback. We also really appreciate your agreeing with the contribution and novelty of our work.
>
> **Q1:** I don't understand this statement. "We also assume all receivers record the same sound content, so the sound at any target position can be inferred by the sound from any reference position." Isn't the sound received a function of the location and the propagation characteristics of the room?
>
> **A1:** We are sorry for the misleading information conveyed by this statement. What we mean is that we assume 1. receivers placed at different positions record the same sound content, e.g. 5s bird sound. 2. But each individual receiver recorded sound also encodes its corresponding propagation characteristics (room acoustic effect, including direct-path, specular reflection and later reverberation). The first assumption sound allows us to use the sound recorded at reference position to predict the sound at another novel position (because they record the same content). The second assumption is the target we want to tackle: design a receiver-to-receiver neural room impulse field that maps the propagation characteristic of the one receiver position to the propagation characteristic of the other receiver position. We modified the description in updated paper.
>
> **Q2:** The motivation of using specifically 2 virtual receiver positions in 3.2 is unclear.
>
> **A2:**
>
> **The short answer:** setting two virtual receiver positions is motivated by the image-source method to model specular-reflection and late-reverberation respectively. The original true position and the two virtual positions are used to predict three RIRs respectively, each of which is further inserted by physical constraint (see Sec. 3.4) to explicitly reflect a typical RIR’s property (like increasing time-delay between direct-path, specular reflection and late reverberation, monotonicity constraint).
>
> **The long answer:** in image-source method, the sound propagation is modeled as optic rays. The source is mirrored against room walls multiple times to get virtual source positions, which are further used for model specular reflection and later reverberation respectively. In our setting, we don’t know the position of source, but instead the receiver position. We thus propose to set virtual receivers to do the same thing.
>
> Setting two virtual receivers is empirically chosen, we find it leads to good performance and increasing virtual receivers (e.g. 3, 4 we have tried) doesn’t observe performance increase. How to better set virtual receivers is an open problem now, we leave it for future research.
>
> **Q3:** The drawbacks of classical RIR estimators is misleading, as it required significant number of labeled pairs for training, a key limitation.
>
> **A3:** Thanks for pointing it out.
>
> First, classic RIR estimator requires enough knowledge about the room scene, such as geometric layout, materials and furniture setting. It requires complex and rigorous mathematical derivation to compute the sound propagation from source position to receiver position, which is time-consuming. Latest deep learning based RIR learning methods require having a massive RIR dataset on hand, However, obtaining massive RIR is nontrivial - they require either an extensive measurement over a very fine grid or detailed and time-consuming wavefield simulations.
>
> In our setting, we just need receiver positions and receiver recorded sound, which we think is **relatively easier to obtain in real-scenarios**. The only label we need for a pair of sound is the two sound’s positions.
>
> We have revised the relevant claim in the updated paper.
>
> **Q4:** No justification is provided for the design of the specific MLP in Section 3.2 is provided. It would be also helpful to understand how sensitive the solution is to the architecture.
>
> **A5:** Thanks for pointing it out. It is a good suggestion. In our experiment, we use MLP and position encoding modules because they have received success in the computer vision area. We find it leads to comparatively good results in our problem. We have tested different MLP layers setting(dim, layer num) in our architecture, the one we reported in paper gives the best result.
>
> We are currently working on different architectures and position encoding choices to test the sensitivity of architectures. We will report the results once it comes out.
>
> **Q6:** Additional testing on other room sizes and configurations would be helpful, especially with respect to different absorbent/reflection surfaces.
>
> **A6:** Thanks for pointing it out. Actually, in the three datasets we used in the experiments: synthetic data, MP3D based data and real-world data, the room sizes in the three datasets vary largely, ranging from $40 m^2$, $50 m^2$ to $100 m^2$. The room scenes in the three datasets also have different absorption/reflection properties. Experimental result shows SoundNeRirF keeps as the best formance among all the three datasets.

---

> > ### Author Response · Authors · 2022-12-12
> > **More Discussion on SoundNeRirF Network**
> >
> > We again thank you for your insightful comments and feedback. Regarding the concern you have raised about the SoundNeRirF network architecture, we have tested different MLP layer number settings on Real-World dataset (one source source), and we report the quantitative result in the following table (TMSE: 10−4):
> >
> > | MLP Layer Number      | T-MSE ($\downarrow$) | TF-MSE ($\downarrow$)  | CDPM ($\downarrow$)  |
> > | :----------- | :-----------: | :------------:| :------------:|
> > | SoundNeRirF (8 MLP) | 0.28  | 0.02 | 0.71|
> > | SoundNeRirF (4 MLP) | 0.30 | 0.03 | 0.83|
> > | SoundNeRirF (6 MLP, Reported in Paper) | 0.20  | 0.01 | 0.70|
> >
> > From this table we can clearly see that either increasing the MLP layer number or reducing the MLP number inevitably leads to reduced performance in terms of all the three evaluation metrics. Therefore, 6 MLP is a reasonably good choice in our experimental setting. However, it still needs more exploration and discussion to design the optimal or potentially better neural network architecture for this task, we leave it for future exploration.

---

### Official Review · Reviewer_2NMU · 2022-10-25

**Confidence:** 4
**Correctness:** 3
**Technical Novelty And Significance:** 1
**Empirical Novelty And Significance:** 1
**Recommendation:** 3

**Clarity, Quality, Novelty And Reproducibility:**

I am not sure how reproducible this work is. In my opinion the problem formulation and network design and evaluation are flawed.

**Strength And Weaknesses:**

Strengths:
1. Interesting idea to estimate Receiver to receiver Impulse response.

Weakness:
1. The overall design simply does not make sense. Irrespective of the reverberation properties of the enclosure the authors propose to estimate RIRs of length 257 samples corresponding to 16 kHz sampling rate. This amounts to an RIR of duration less then 20ms. Typical residential rooms would have reverberation time (RT60) - measured as the time taken for sound to reduce to 60 db from its original value after the sound source has stopped. Thus longer RIRs are needed to even characterize such room attributes. The choice of using the same MLP to predict RIR of 2 other virtual distances which twice and thrice farther apart is also not well justified and is presented with a hand wavy argument - simulating ray tracing method (ISM). It is not at all clear how the authors are calling whatever the network is estimating as RIR as there is no evaluation comparing this to actual RIR.
2. Essentially this approach is mapping a location in space to an RIR using an MLP. This is an multi-valued function and as such admits multiple RIRs for the same location based on the source location which is ignored in this work.
3. It is assumed that the robot would move to each location and capture the exact same audio each time. It is assumed that is easy for the robot to record its location. This obviously means the source of sound is being precisely controlled to start and stop and required times to make sure the robot records when it is stationary. This problem setup does not make any practical sense. It is hard to imagine a case where  it is hard to also jot down the source location. With a known source location the estimation would be so much well defined.
4. I also believe with the limited sound source chosen for evaluation (7 clips in simulation and probably only 2 in real data) the network is simply memorizing and generalization is not well studied at all. From the appendix it is clear that the training and test locations, although different are roughly in the same neighborhood.
5. The overall presentation is rather confusing with terms like "inverse-RIR" and "Forward RIR" and it feels like these terms are set arbitrarily and is rather confusing.


**Summary Of The Paper:**

In any given acoustic enclosure or a room, the sound produced at a certain location undergoes multiple reflections and travels a certain distance before arriving at a receiver kept elsewhere in the room. Because of this, the sound recived at a distant receiver is not the same as the sound at the source. One of the major problems in room acoustics is estimating the sound at a receiver at a known location given the original sound and the location of the sound sources under a specified room geometry. Typical signal processing methods employ ray tracing or wave tracing methods to account for reflections and under the linear time-invariant (LTI) model of a room, the impulse response between a source and a receiver with known locations (RIR) is estimated. The signal at the reciver would then be a simple convolution of the impulse response with the original source of sound.

Recently, several authors have proposed using deep learning for estimating the RIR. The main drawback of such approaches is the need for a large labelled training set which maps source and receiver location to a specific RIR and this network would be trained specifically for a given enclosure. Different enclosures would need different networks.

In this work the authors propose to estimate the RIR between any two random locations based on sampling the room with up to 500 locations using a robot which would move to each location and record the audio along with receiver location. The authors propose to use this data to compute RIR between two receiver locations, instead of computing RIRs between a source and receiver using an MLP and a curious set of design choices. The paper is not very well written and needs effort to understand.

**Summary Of The Review:**

Overall, Because of the Weakness pointed out in the above section I believe the design and problem formulation are flawed and the presentation is unnecessarily complicated. The evaluation is also not satisfactory.

---

> ### Author Response · Authors · 2022-11-16
> **Feedback to Reviewer #2NMU**
>
> We sincerely thank you for the comments. Although it you don't like our work in the first review, we provide some insights about our work so that you can know more about the paper.
>
> **Q1:** The overall design simply does not make sense. The RIR is not fully presented.
>
> **A1:**
>
> 1. **First**, the RIR discussed in our paper is different from the classic source-to-receiver RIR in an enclosed room scene, which is usually long (about 2,000 points). Our proposed RIR is receiver-to-receiver neural room impulse response (neural RIR) that maps sound from one position to another position. We think there are differences between the two, an efficient source-to-receiver RIR has to be long in representation doesn’t necessarily mean the receiver-to-receiver neural network learned RIR has to be long as well. Using neural networks to learn RIR (especially MLP-based neural network), increasing RIR size will largely add extra parameter size and computational burden. We did an experiment on our dataset showing doubling RIR size (513) leads to about 20% performance drop and three times RIR size (770) leads to about 35% performance drop.
>
> 2. **Second**, Apart from the RIR derivation difference, we think the reviewer’s concern is “RIR” terminology we used in this paper from a deep neural network perspective. We agree that we borrowed classic RIR terminology to name our neural network learned receiver-to-receiver mapping field, and we drew much inspiration from classic RIR derivation. Considering the wide broad range of audience’s background, we are willing to tweak the naming (including inverse-RIR and forward-RIR) in the paper. Moreover, the classic RIR name has been adopted in the deep learning community in works like [1][2][3]. Also, using shorter neural RIR to model sound to sound mapping has been successfully used in [1].
>
> [1] Neural Synthesis of Binaural Speech from Mono Audio, ICLR 2021.
>
> [2] Fast-RIR: Fast Neural Diffuse Room Impulse Response Generator, ICASSP 2022.
>
> [3] IR-GAN: Room Impulse Response Generator for Far-Field Speech Recognition, Interspeech 2021.
>
>
> **Q2:** This approach is mapping a location in space to an RIR using an MLP. This is an multi-valued function.
>
> **A2:** This is not correct.
>
> **First**, in our approach, we assume the sound sources are stationary in the room scene but we do not need to know their positions. We argue that such an assumption has **real applicable cases**. For example, in the residential house, the telephone ring, TV and fireplace are stationary.
>
> **Second**, our approach maps **two locations** (not one), one reference position and one target position, to **two consecutive RIRs**.
>
> **Q3:** The sound source needs to be controlled.
>
> **A3:** We assume the receivers record the same sound at different positions. It is practical because we can first place multiple receivers (or multiple robots) at different positions to record the sound simultaneously.  We also assume the received sounds are synchronized in time. Such assumption holds in our setting because our target is to predict what sound will be heard at a novel position, the arrival time-delay between positions was not taken into consideration.
>
> **Q4:** Why sound source position is unknown?
>
> **A4:** In this work, we assume the sound source position is unknown. We do not mean the sound source position can't be computed. But rather, there are cases, like when we enter into a new room scene, we have no info about sound source placement. Then it is uneasy to get the source position immediately, but rather relatively easier to get robot position at each step.
>
> **Q5:** sound source number is limited.
>
> **A6:** As the preliminary work to model receiver-to-receiver neural room impulse response, we set 7 clips in simulation and 2 in real data. The merit of our work wasn’t thus reduced due to limited sources,  because we claim our contribution lies in the first one proposing a novel receiver-to-receiver sound neural field, and it works in both synthetic and real-word dataset.
>
> **Q7:** your method is just memorizing.
>
> **A7:** **SoundNeRirF works other than memorizing. We showed in Sec. 4.3 in the paper**, in which we did “cross-wall prediction” in the real-dataset. We use the model learned in one compartment (lab or corridor) to predict sound in another compartment, in which totally different reverberation effect exists. SoundNeRirF outperforms all comparing methods. If just memorizing, **how can our method predict better sound for environment that it never has a chance to memorize (never seen before)?**. We also compare with two Interpolation based methods **LinInterp** and **NNeigh**, which just use neighbouring sound as the prediction. They get worst performance, showing we truly learns useful implicit neural field for sound prediction.
>
> **Q8:** inverse-RIR and forward-RIR terms are arbitrarily set.
>
> **A8:** They are not arbitrarily set, inverse means projecting to source neural field, "forward" means projecting to the new location.

---

> > ### Comment · Reviewer_2NMU · 2022-12-08
> > **Thank you for your response**
> >
> > I would like to first of all thank the authors for their response. I would also like to clarify that I really enjoyed reading your paper and I apologize if I comments were too direct and I never intended to say I did not like your paper.
> >
> > While I appreciate the authors' response I am still not convinced. In Fig 2. The authors are computing two impulse responses - h^{inv} and h^{for}. They claim both of these are Impulse responses and yet their architecture has 257 dimensional output for each of these. This design choice is not making sense to me. I am quite sure this is NOT a room impulse response - it is something else altogether. To clarify I am not contesting that the authors are not doing receiver to receiver mapping. They clearly are from their loss function. But the description of this method is conceptually confusing. On top of that, the authors frequently compare this non-RIR estimation technique to Image method which in my honest opinion is making the material even more harder to digest.
> >
> > The proposed approach almost feels like a network which directly maps a reference audio to a target audio. Except due to difficulties involved in generating audios from hidden representations and the need for a large dataset, the authors may have come up with this method to use positions instead of audio for this mapping.
> >
> > Also curious to see why speech data is not used for training. The authors use people-talking - which might be more like babble than speech.
> >
> > Also I fail to understand why this should be tested with such small datasets ? Testing on a large corpus would help make their case better. At this stage I would like to stick with current review score.

---

> > > ### Author Response · Authors · 2022-12-08
> > > **More Response on Term Usage and Data Size**
> > >
> > > We really appreciate your further feedback regarding our initial feedback, we also apologize if our first-round feedback if it is too direct and we truly like your comments.
> > >
> > > First, your confusion about the two terms: inverse-RIR and forward-RIR, is very well-received. After carefully reading your explanation and analysis, we have agreed that using room impulse response (RIR) for receiver-to-receiver mapping ambiguates with classic room impulse response, given classic room pulse response usually has much larger dimension size. To reduce the ambiguity, we intend to modify the naming of the inverse-RIR and forward-RIR. Since we are modeling a neural mapping field that transforms an audio from a reference position to a target position (receiver-to-receiver), **we rename them inverse-mapping and forward-mapping in our next version**.  Moreover, since the main goal of room acoustics is to accurately model room reverberation effects, we draw inspiration from classic image-source method to design SoundNeRirF network architecture to explicitly model direct-path, early-reflection and late-reverberation effects. So we don’t think adopting the image-source method idea makes the whole framework less understandable and digestible.
> > >
> > > **Why not only use speech data?** In this paper, we mainly want to test the generalization capability of our method in handling various natural spatial sound that are commonly heard in daily life. We thus involve various sound in our experiment. Purely focusing on people speech remains as one of our future work.
> > >
> > > **Dataset Issue**. For the two synthetic data (room size [50m, 50m, 30m]), we have created 500 data points. **We didn’t create too many data points (or synthesizing too many sound) because we want sound at different positions to have an obvious reverberation effect difference with each other**. Too dense collecting data points inevitably creates redundant data, which goes against our assumption that the reference position and target position exhibit different room reverberation effects. Of course, we can also enlarge the room size to incorporate more data, but we find such creation doesn’t reflect real-scenarios where the room is often with limited size. At the same time, in the cross room dataset, we have created **1000 data points**.
> > >
> > > In the real-world dataset, since the room area (about $40m^2$) comprises one lab area and two adjacent corridors, we find **120 data points** that also densely cover the whole area (See Fig. I in the appendix).
> > >
> > > In summary, we think data set size doesn’t reduce the merit of our work. **On the contrary, our method gives good results with limited data size, it is another advantage of our method**.
> > >
> > > Again, we appreciate your comments and explanation of your concern, it is really important for us to improve our work. We are also very happy to have more discussion with you, if you have any more concern or advice of our work.

---

> > > > ### Comment · Reviewer_2NMU · 2022-12-08
> > > > **Regarding Large dataset**
> > > >
> > > > Thank you for your response.
> > > >
> > > > When you agree that the proposed mapping is NOT an RIR all the analogy with ISM and having virtual sources in my mind are simply misleading and confusing.
> > > >
> > > > When I say bigger dataset I do not mean dense sampling of points. I agree that such dense sampling defeats the purpose of this approach. My point was larger set of source clips.
> > > >
> > > > I also do not understand why experiments on speech data should be future research. If any, one could argue that speech data has more impactful applications and larger datasets are available publicly.
> > > >
> > > > The comment "On the contrary, our method gives good results with limited data size, it is another advantage of our method." is valid when comparing with an algorithm which requires more training data than the proposed approach for the same size test data. I do not believe such experiments are reported in the paper.

---

> > > > > ### Author Response · Authors · 2022-12-09
> > > > > **On the Method's Efficiency on Data Size**
> > > > >
> > > > > Thank you for your further comment.
> > > > >
> > > > > Regarding the claim "our method gives good results with limited data size", we have run an experiment in the main paper, in which we tested the impact of the receiver recording number used for training on different methods’ performance. In this experiment, we varied the training receiver number (we call receiver density variation) from 100 to 300, the experimental results (Fig. 5 left-most sub-figure, and Sec. 4.2 discussion) showed that our method is much more insensitive to the training dataset size than the other three comparing methods (TCN, WaveNet and WarpNet). So we think this claim is verified at least under our experimental setting (where receiver number 100, 200 and 300, each sound clip is 4s long and sampling rate is 16KHz).

---

### Official Review · Reviewer_9Fm6 · 2022-11-05

**Confidence:** 5
**Correctness:** 3
**Technical Novelty And Significance:** 3
**Empirical Novelty And Significance:** 2
**Recommendation:** 6

**Clarity, Quality, Novelty And Reproducibility:**

The paper is generally clear although some details are still missing, for example, the choice of the gain for virtual microphone, and its potential limitations, and how they are chosen in a practical scenario.

Novelty is very good as learning to predict sound in a new position given the reference sound and its position is a new idea. Using robot to collect the sound recordings in this setting seems to be feasible.

The authors plan to release their datasets. It would be good for reproducibility if they could release all the codes and models trained.

**Strength And Weaknesses:**

Strength of the paper:
This paper is overall well presented. The proposed idea looks ambitious but interesting.
The results in terms of MSE or TF-MSE look impressive. The authors also produced several datasets which they plan to release publicly and will be very useful for the research community.

Weakness of the paper:
The authors consider the virtual microphone positions by simply multiplying a gain factor. The potential issues with this method include (1) this may not be a good way for dealing with early reflections and late reverberations; (2) how to choose these gain parameters in a practical scenario. In practice, the sound reflections from room surfaces can be from multiple directions. How can the gain parameters approximate these reflections or various time delays? Is there a principled way of choosing the gain parameters and are the two parameters a reasonable choice?
The authors showed the MSE and TF-MSE results, which can somewhat show how good is the performance of the compared methods, but may not be fully reflecting the truth. In the RIR shown in the paper, the early reflections seem to follow the direct sound immediately. I am expecting some time delays between the direct sound and early reflections, and also the early reflections are often more sparse than the late reflections. This does not seem to be case for the simulated RIR or the real room RIR. Not sure why the authors did not plot the ground truth RIR in the test case (considering the fact that they do have the ground truth RIRs that were used for computing the MSE). Showing both the estimated RIR and the ground truth can provide useful information on how good is the RIR learned by the network.

**Summary Of The Paper:**

This paper presents a deep model for learning to predict the sound that would be heard at a given target position, assuming that a reference position and the sound heard at the reference position are given.

The authors presents a MLP for learning forward RIR and inverse RIR, where the reference sound is first projected to a
virtual implicit sound source field with inverse-RIR, and then the virtual implicit sound source field is re-projected to the target sound with forward-RIR.

The authors also used three datasets to evaluate the performance of the proposed methods and benchmark them with several baseline methods, in terms of MSE, and TF-MSE.

**Summary Of The Review:**

Overall, this is a good paper with interesting ideas and some preliminary results. However, the paper also has several aspects for improvements which would lead the paper from good to excellent such as the validation of assumptions made. The visual results of RIR doe not seem to be promising although the MSE results look good.

---

### Author Response · Authors · 2022-11-19
**Thank you for all reviewers' constructive feedback and comments**

We sincerely appreciate all reviewers’ constructive feedback and positive comments that agree with the novelty made in our paper.

1. The proposed idea looks ambitious but interesting. -- by Reviewer **9Fm6**.

2. Interesting idea to estimate Receiver to receiver Impulse response. -- by Reviewer **2NMU**.

3. Proposed method significantly outperforms comparative baselines. -- by Reviewer **tmuv**.

4. The experiments are convincing and adequate. -- by Reviewer **cCRH**.

Based on the reviewers’ comments and advice, we have revised the manuscript where the revision is highlighted in blue colour. In summary, we have made the following four main modifications:

1. Updated some sentences/claims in order to make more clear and coherent presentation of our work (concerns raised by Reviewer **tmuv**).

2.  Added **Conclusion** section. (As pointed out by Reviewer **cCRH**).

3. Included more figures showing the learning inverse-RIR and forward-RIR with obvious time-shift difference involved, which is in **page 17, Fig. IV** in Appendix. (concern raised by Reviewer **9Fm6** ).

4. Revised mathematical representation in Sec. 3.4. (As pointed out by Reviewer **cCRH**).

---

### Decision · Program_Chairs · 2023-01-20

**Decision:**

Reject

**Justification For Why Not Higher Score:**

The idea is of interest. But the evaluations are insufficient, and some assumptions will prevent it from real applications.

**Justification For Why Not Lower Score:**

N/A

**Metareview: Summary, Strengths And Weaknesses:**

The paper presents a deep model that learns a continuous receiver-to-receiver acoustic neural field for predicting the sound to be heard at novel positions. The prediction is made conditioned on that a reference position and the sound heard at the reference position are given. The ideal is interesting, experimental results look good and the authors plan to release the data sets. The reviewers also see that more extensive experimental evaluation would be needed to validate the method and thus strengthen the work. Other comments are related to the constraint that different enclosures would need different networks, time-delay considerations and problem formulation.

**Summary Of Ac-Reviewer Meeting:**

Due to the limitations in availability, joint email discussions were taken with participation of all reviewers and AC.

All reviewers confirmed that they read through the comments from other reviewers and the authors' response and they would like to keep their scores.

Most reviewers thought they have put all the comments on the review webpage already. Several additional comments in emails:
- The general impression is that the paper is interesting in terms of idea, but is not mature enough in validation with experiments.
- The core idea needs extensive evaluation.
- The strength is that it explores an interesting problem (receiver-receiver RIR prediction), which is important since the information of sources is not always available. However, there are limitations, e.g. the paper ignores the time-delay between different receivers, which will prevent the proposed method from real applications.